# UNIQA: UNIFIED VISION-LANGUAGE PRE-TRAINING FOR IMAGE QUALITY AND AESTHETIC ASSESSMENT

## ABSTRACT

Image Quality Assessment (IQA) and Image Aesthetic Assessment (IAA) aim to simulate human subjective perception of image visual quality and aesthetic appeal. Despite distinct learning objectives, they have underlying interconnectedness due to consistent human assessment perception. Existing unified methods typically combine datasets of two tasks for regression training directly, which fail to learn mutually beneficial representations shared by both tasks explicitly. To confront this challenge, we propose **Uni**fied vision-language pre-training of **Q**uality and **A**esthetics (**UniQA**), to extract useful and common representations from two tasks, thereby benefiting them simultaneously. Unfortunately, the lack of text in the IQA datasets and the textual noise in the IAA datasets pose severe challenges for multimodal pre-training. To address this, we (1) utilize multimodal large language models (MLLMs) to generate high-quality text descriptions; (2) use the generated text for IAA as metadata to purify noisy IAA data. To effectively adapt the pre-trained UniQA to downstream tasks, we further propose a lightweight adapter that utilizes versatile cues to fully exploit the extensive knowledge of the pre-trained model. Extensive experiments show that our approach achieves state-of-the-art performance on both IQA and IAA tasks, while also demonstrating exceptional few-label image assessment capabilities.

## 1 INTRODUCTION

Image Quality Assessment (IQA)[1] and Image Aesthetic Assessment (IAA) aim to measure the perceived quality and beauty of an image. They find broad applications in many scenarios, such as guiding individuals in image photography and editing, and serving as tools for image dehazing model (Zhao et al., 2021). Consequently, huge efforts (Su et al., 2020; Ke et al., 2021; He et al., 2022) have been devoted to establishing effective IQA and IAA models.

IQA and IAA concentrate on distinct aspects of image assessment, with IQA primarily focusing on the distortion level of the image, while IAA is oriented towards evaluating the aesthetic appeal of the image. Despite their differences, IQA and IAA have underlying commonality: **simulating human subjective perceptions of images.** Specifically, in human subjective evaluation of images, quality and aesthetics exhibit a mutual influence, such that high-quality images are more likely to possess a higher aesthetic appeal compared to their low-quality counterparts. Thus, the learning process for both tasks not only acquires features unique to themselves but also involves the learning of task-agnostic common representations. This commonality sparks an idea:

> *Can we develop a foundational model with robust visual assessment perceptions consistent with human to benefit both IQA and IAA tasks?*

Although previous works (*e.g.*, MUSIQ (Ke et al., 2021)) can be applied to IQA and IAA tasks indiscriminately, they cannot exploit beneficial representations from another task. Wu et al. (2023b) and Zhang et al. (2023a) also find the similarities of two tasks and attempt to tackle them with unified architecture and training. However, they typically unify datasets of two tasks for regression training directly, which cannot explicitly learn the task-shared representations, restricting the extraction of mutual benefits. In this paper, we propose the **Uni**fied pre-training of **Q**uality and **A**esthetics

---

[1]The IQA in this work refers to the no-reference image quality assessment.

Figure 1: The overview of our method. We leverage MLLMs to generate quality- and aesthetics-related descriptions (Step 1) and utilize the generated data to refine authentic noisy data (Step 2). We conduct unified pre-training to obtain UniQA (Step 3), which can be flexibly applied to both IQA and IAA tasks with a lightweight adapter (Step 4).

(UniQA) to extract mutually beneficial and effective representations for both tasks. Then, the pre-trained UniQA can be flexibly applied to IQA and IAA datasets.

To achieve unified pre-training, a straightforward solution involves consolidating all IQA and IAA datasets and then training the model to regress towards the mean opinion scores (MOS) annotated by humans. However, existing datasets show variations in perceptual scales due to differences in subjective testing methodologies (Zhang et al., 2021a). As a result, this training strategy makes the model develop a score bias toward larger scale datasets. Moreover, it may not effectively capture the unique characteristics of IQA and IAA, as the MOS labels cannot be explicitly interpreted. To this end, we propose to use **text descriptions** as a bridge to integrate the two tasks, leveraging the rich and fine-grained semantics inherent in text to provide more auxiliary information.

However, existing IQA datasets typically have images only and lack text descriptions. While current IAA datasets (Ghosal et al., 2019) include text data provided by humans, they often contain considerable textual noise irrelevant for aesthetic assessment. Therefore, a top priority is determining how to acquire high-quality image-text data for both IQA and IAA tasks. Recently, multimodal large language models (MLLMs) (Liu et al., 2023b; Zhu et al., 2023; Lin et al., 2023; Bai et al., 2023) have demonstrated outstanding capabilities in image understanding, which can generate reasonable responses based on images and user instructions. Inspired by this, we propose utilizing MLLMs with tailored prompts to generate quality- and aesthetics-related descriptions for the IQA and IAA datasets, respectively (Step 1 of Figure 1). As observed in Figure 2, this approach provides a comprehensive and precise depiction of image quality and aesthetics. Furthermore, we utilize these generated high-quality aesthetics-related descriptions as metadata to refine the raw aesthetic caption dataset (Step 2 of Figure 1). Finally, we unify the generated and refined image-text datasets to conduct vision-language contrast pre-training (Step 3 of Figure 1). This results in the pre-trained UniQA with a powerful multimodal image assessment perception.

After pre-training on image-text pairs, we propose a lightweight adapter, namely the Multi-Cue Integration Adapter, to fine-tune the specific dataset of two tasks (Step 4 of Figure 1). This adapter uses versatile cues related to image assessment to prompt the pre-trained UniQA, adeptly extracting useful knowledge and comprehensively assessing the image. With much fewer tunable parameters compared to previous IQA and IAA models, our model outperforms them on both tasks. More surprisingly, benefiting from the powerful representations learned by pre-training, our method achieves impressive results on few-label IQA, *e.g.*, achieving the SRCC values of 0.828 (vs. 0.760 on CLIVE of GRepQ (Srinath et al., 2024)) and 0.844 (vs. 0.812 on KonIQ of GRepQ).

Our contributions can be summarized as follows:

- With the assistance of MLLMs, we construct a high-quality image-text dataset about image quality and aesthetics. Through pre-training on this dataset, we develop UniQA, which

effectively learns a general perception of image assessment, promoting the effective and efficient learning of both IQA and IAA tasks.

- We propose a novel Multi-Cue Integration Adapter, which integrates various assessment-related cues to fully exploit the extensive knowledge of the pre-trained model with minimal additional parameters.

- Extensive experiments show that our method achieves SOTA performance across multiple IQA and IAA datasets. Benefiting from the rich representations learned through pre-training, UniQA also demonstrates exceptional few-label image assessment capabilities.

## 2 RELATED WORK

**Image Quality Assessment.** The rapid development of deep learning has sparked significant interest in their application for IQA. Many researchers utilize CNN to solve the IQA problem with various effective techniques, including multi-level feature aggregation (Li et al., 2018), adaptive quality prediction (Su et al., 2020), and patch-to-picture learning (Ying et al., 2020), order learning (Shin et al., 2024) and unsupervised learning (Saha et al., 2023). Recently, transformer-based IQA methods (Ke et al., 2021; Zhu et al., 2021; Qin et al., 2023; Xu et al., 2024; Yu et al., 2024) show promising results in the IQA field, which can compensate for the non-local representation ability of CNN. Despite these impressive breakthroughs, these methods often transfer models pre-trained on classification datasets, such as ImageNet (Deng et al., 2009), to IQA tasks, which may be suboptimal (Li et al., 2023d). Q-Align (Wu et al., 2023b) attempts to jointly perform IQA and IAA tasks, but it uses a language model with a huge number of parameters and does not explicitly extract features of the two tasks through pre-training. Our method can learn more effective representations through joint pre-training on quality-aesthetics image-text data, providing benefits for IQA tasks.

**Image Aesthetic Assessment.** With the advent of deep learning, IAA methods have evolved from hand-crafted feature extraction (Datta et al., 2006; Ke et al., 2006; Nishiyama et al., 2011; Sun et al., 2009) to end-to-end feature learning, marking significant advancements in the IAA domain. Various techniques have been developed to boost IAA task, including local and global feature integration (Lu et al., 2015; Hou et al., 2020; Shi et al., 2024; Huang et al., 2024a; He et al., 2023b), graph network (She et al., 2021; Duan et al., 2022) and theme-aware learning (Li et al., 2023c; He et al., 2022). Recently, there has been an emergence of multimodal IAA methods (Zhang et al., 2020; Zhou et al., 2016; Zhang et al., 2021b; Nie et al., 2023; Huang et al., 2024b) that incorporate text as auxiliary supervision. However, these methods necessitate the use of text during inference, limiting their flexible application since text is often not easily available. Our method overcomes this limitation by conducting vision-language pre-training firstly to learn effective representation. The pre-trained model can be flexibly applied to the IAA field using only images.

**Vision-Language Models.** Vision-Language Models (VLMs) (Radford et al., 2021; Jia et al., 2021; Yao et al., 2021; Yu et al., 2022; Sun et al., 2023) introduce the contrastive learning strategy to acquire image-text correspondences from large-scale image-text pairs. VLMs have exhibited promising results across multiple tasks, including IQA (Wang et al., 2023; Zhang et al., 2023b) and IAA (Hentschel et al., 2022; Sheng et al., 2023). Recently, the Multimodal Large Language Models (MLLMs) have garnered increasing research interest, exhibiting remarkable prowess in comprehending image content and reasoning through complex instructions (Liu et al., 2023b; Zhu et al., 2023; Li et al., 2023a; Ye et al., 2023; Bai et al., 2023). Most existing MLLMs achieve this by integrating image features with LLM tokens, subsequently fine-tuning the LLM via multimodal instruction tuning. During inference, MLLMs can reason with given images and user instructions, generating text responses by leveraging world knowledge learned during pre-training.

## 3 UNIQA: MLLMS-ASSISTED UNIFIED PRE-TRAINING

In this section, we first present some preliminaries of related models (Section 3.1). We then describe the process of constructing a unified image-text dataset about quality and aesthetics, with the assistance of MLLMs (Section 3.2 and 3.3). We use this dataset to pre-train the vision-language model (Section 3.4) to construct our UniQA.

**Quality-related captioning**
**MOS of image**: 7.36 → **Text level**: good
**Prompt**: What you are given is a good quality image. Evaluate image quality based on factors such as *sharpness, color balance, and noise level*.
**Output**: The image is of a living room with a fireplace, a couch, a chair, and a coffee table. The room is well-lit, and the fireplace adds warmth to the space. The image is of high quality, with good resolution, minimal noise, and appropriate brightness levels.

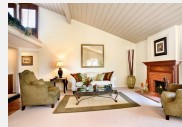

**Aesthetics-related captioning**
**MOS of image**: 4.86 → **Text level**: fair
**Prompt**: What you get is an image of fair aesthetics. Evaluate image aesthetics based on factors such as *content, color, lighting, and composition*.
**Output**: The image presents a medium aesthetic quality, with a focus on the snow-covered highway and the mountain range in the background. The overall composition of the image is balanced, with highways and mountains as the main subjects, and the cars and trees providing additional interest and context.

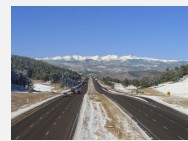

Figure 2: Generated quality- and aesthetics-related captions via MLLMs. The red text refers to MOS-based text guidance. The orange text highlights the quality- and aesthetics-related text.

### 3.1 PRELIMINARIES

Vision-language pre-training aims to achieve comprehensive cross-modality understanding by training on web-scale image-text datasets. Benefiting from this large-scale pre-training, CLIP (Radford et al., 2021), a prominent VLM, has demonstrated great promise to assist a broad scope of vision tasks. Specifically, CLIP comprises an image encoder $f$ and a text encoder $g$, both jointly trained to establish a shared latent space for image and text through contrastive learning.

Given a batch of $N$ paired images and texts $\{x_I^i, x_T^i\}_{i=1}^N$, CLIP extracts image features $\boldsymbol{I} = \{f(x_I^i)\}_{i=1}^N$ and text features $\boldsymbol{T} = \{g(x_T^i)\}_{i=1}^N$ with corresponding encoders. During pre-training, CLIP seeks to maximize the cosine similarity of paired image and text features, while minimizing the similarity of unmatched pairs. The contrastive learning objective can be formulated as:

$$\mathcal{L}_{\text{image}} = -\mathbb{E}_{I_i \sim \boldsymbol{I}} \left[ \log \frac{\exp(I_i^\top T_i / \tau)}{\sum_{j=1}^N \exp(I_i^\top T_j / \tau)} \right]$$
$$\mathcal{L}_{\text{text}} = -\mathbb{E}_{T_i \sim \boldsymbol{T}} \left[ \log \frac{\exp(T_i^\top I_i / \tau)}{\sum_{j=1}^N \exp(T_i^\top I_j / \tau)} \right] \tag{1}$$

where the $I_i$ and $T_i$ are the $i$-th features in the batch, and $\tau$ is the temperature parameter. The final contrastive learning loss can be obtained by taking the average: $\mathcal{L} = (\mathcal{L}_{\text{image}} + \mathcal{L}_{\text{text}})/2$. With this training strategy, CLIP can generate aligned features in latent space for paired image-text samples.

### 3.2 QUALITY- AND AESTHETICS-RELATED CAPTIONING

In order to achieve vision-language pre-training in the field of image assessment, we need to generate text for IQA and IAA datasets since IQA datasets lack text and IAA datasets contain noisy text. Recently, MLLMs have shown advanced performance, so we can use them to generate high-quality textual data for images. Previous studies (Wu et al., 2023a; Huang et al., 2024c) have highlighted that it is challenging for MLLMs to directly and accurately perceive the quality and aesthetics of input images, often resulting in positively skewed expressions and strong hallucinations (see examples in Appendix D). Thus, to obtain correct and detailed descriptions about quality and aesthetics, as shown in Figure 2, we design ***MOS-guided task-specific prompts*** to instruct MLLMs:

$$Y_t \sim M_T(x_I, P_t | G). \tag{2}$$

where $M_T$ denotes the used MLLM, $G$ is the MOS-based text guidance, $P_t$ is the task-specific prompt, $Y_t$ represents the generate caption. To obtain $G$, we divide images into 5 levels based on MOS, *i.e.*, {bad, poor, fair, good, perfect} (Ghadiyaram & Bovik, 2015; Sheikh et al., 2006; Zhang et al., 2023b). If an image's MOS ranks in the top 20% of the score range, its level is assigned to perfect. This approach harmonizes IQA and IAA datasets with different MOS scales, alleviating the MOS biases of different datasets (Zhang et al., 2021a). Additionally, $P_t$ is customized for IQA ($P_{IQA}$) and IAA ($P_{IAA}$) tasks, respectively. As shown in Figure 2, $P_{IQA}$ involves *sharpness, color balance, and noise level* (Chandler, 2013), while $P_{IAA}$ includes *content, color, lighting, and composition* (Deng et al., 2017). With these designs, $M_T$ is guided towards image assessment

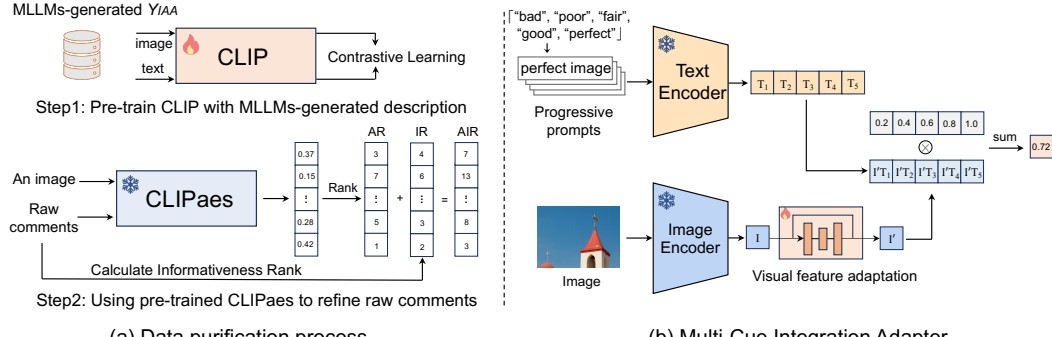

Figure 3: (a) Data purification process: we pre-train CLIP using generated aesthetic captions data $Y_{IAA}$ and then use the pre-trained $\text{CLIP}_{aes}$ to purify data. (b) The proposed adapter: we employ progressive prompts, {bad, poor, fair, good, perfect} with "image", to prompt the frozen UniQA and a lightweight trainable module to adjust visual features.

and we can obtain generated caption datasets $Y_{IQA}$ and $Y_{IAA}$ for IQA and IAA tasks, respectively. For simplicity and cost-effectiveness, we use open-source LLaVa (Liu et al., 2023a) as the captioner. We also experiment with the effects of different MLLMs on model performance (Table 6).

## 3.3 DATA PURIFICATION STRATEGY

In addition to the generated aesthetic captions $Y_{IAA}$, there are also IAA datasets with captions commented by humans (Ghosal et al., 2019), which directly reflect human aesthetic feelings. Incorporating comments from various people can offer a more comprehensive description of image aesthetics. However, while enhancing text diversity, it may introduce noise to the data, as individuals may provide comments unrelated to image aesthetics. To address this issue, we propose a novel data purification strategy to refine raw captions in the original dataset. This process is illustrated in Figure 3(a).

Specifically, we introduce ***Aesthetics-relevance and Informativeness Rank (AIR)*** to measure the quality of text corresponding to an image. The AIR consists of Aesthetics-relevance Rank (AR) and Informativeness Rank (IR). To obtain AR, we first pre-train a CLIP model with generated aesthetic data $Y_{IAA}$ to get an aesthetics-aware CLIP model, denoted as $\text{CLIP}_{aes}$. Then, we employ it to measure the aesthetics relevance score ($s_A$) for an image-text pair. Given an image with $n$ captions, AR can be defined as:

$$\text{AR} = \text{Rank}(s_A^1 \cdots s_A^n), \quad s_A^i = \text{CLIP}_{aes}(x_I, x_T^i), \tag{3}$$

where $s_A^i$ represents the aesthetics relevance score between the $i$-th caption $x_T^i$ and its corresponding image $x_I$. Note that AR consists of *long integers* that represent the rank of a caption after sorting by $s_A$. To obtain IR, we simply utilize the sentence length as informativeness score ($s_I$) to measure the informativeness of text. Accordingly, for an image with $n$ textual captions, IR can be expressed as:

$$\text{IR} = \text{Rank}(s_I^1, \cdots, s_I^n), \quad s_I^i = \text{Length}(x_T^i), \tag{4}$$

where $\text{Length}(\cdot)$ is able to output the length of an input sentence. As a result, AIR between an image and $n$ captions is:

$$\text{AIR} = \text{Rank}((\text{AR}^1 + \text{IR}^1), \cdots, (\text{AR}^n + \text{IR}^n)). \tag{5}$$

We select captions with Top-K ranking AIR to construct a high-quality aesthetic caption dataset, denoted as $Y_{IAA}^+$. This strategy ensures the preservation of text that is both related to aesthetic perception and rich in information, thereby enhancing the quality and richness of the raw dataset.

## 3.4 UNIFIED VISION-LANGUAGE PRE-TRAINING

So far, we have gotten a high-quality image-text dataset about quality and aesthetics, $Y = Y_{IQA} \cup Y_{IAA} \cup Y_{IAA}^+$. Based on it, we pre-train CLIP using Equation 1 to obtain our UniQA. In this way, the model learns general perceptions of image quality and aesthetics, which can provide potent assessment priors and thus can be effectively applied to both IQA and IAA tasks.

## 4 ADAPTING VISION-LANGUAGE MODEL FOR IQA AND IAA

The pre-trained UniQA contains extensive perception information, which can facilitate downstream assessment tasks in a zero-shot or supervised manner. In this section, we further propose a meticulously designed adapter (Section 4.1) and prompt ensemble strategy (Section 4.2) to enhance the model's performance.

### 4.1 MULTI-CUE INTEGRATION ADAPTER

During pre-training, the model aligns image and assessment-related captions, empowering it with strong comprehension of image quality and aesthetics. With this foundation model, we can slightly adjust the visual features, efficiently adapting it to score-based image assessment tasks. To this end, we introduce a lightweight adapter, namely the ***Multi-Cue Integration Adapter***, to adapt visual features and inject rich cues for fine-tuning downstream tasks. The adapter consists of two key processes: visual feature adaptation and multi-cue integration prediction.

**Visual Feature Adaptation.** We add a learnable residual module following the pre-trained image encoder to adjust the visual features so as to adapt to specific assessment datasets. We optimize this module while keeping the image and text backbones frozen, enabling parameter-efficient tuning. The structure of the adapter is illustrated in Figure 3(b). Let $I$ denote the image features extracted from the frozen image encoder, the visual feature adaptation process can be expressed as:

$$I' = \text{Normalize}(\text{Adapter}(I) + I) \tag{6}$$

where the $\text{Adapter}(\cdot)$ consists of two fully connected layers with a ReLU activation in between, and $I'$ represents the adapted visual features.

**Multi-cue Integration Prediction.** A straightforward approach to incorporating the CLIP model into perception assessment is to utilize the "`good image`" as an anchor and take the cosine similarity between the text anchor and a given image as the assessment score. However, this method shows two shortcomings: (1) using the absolute value of similarity as the perception score may not be optimal because it only reflects the semantic similarity between images and texts (Wu et al., 2023a; Wang et al., 2023); (2) a single prompt may not fully leverage the extensive knowledge of the pre-trained model. Thus, we propose to utilize versatile cues to comprehensively explore the power of the pre-trained UniQA and convert absolute similarity scores into relative values for weighting.

Specifically, we utilize the prompt template "{level} image" and five text levels (`bad`, `poor`, `fair`, `good`, `perfect`), *i.e.*, "*Multi-cue*", to construct prompts. Next, we calculate the cosine similarity between the normalized text features $\{T_i\}_{i=1}^{5}$ of five prompts and adapted visual features $I'$, and then use the $\text{Softmax}(\cdot)$ to obtain the related value of five image-text correspondence. These related values will weight the predefined score levels to get the final assessment score. This process can be formulated as follows:

$$q = \sum_{i=1}^{5} \frac{c_i \exp(I'^\top T_i/\tau)}{\sum_{j=1}^{5} \exp(I'^\top T_i/\tau)}, \tag{7}$$

where $\{c_i\}_{i=1}^{5}$ are scores of text levels with progressive values that are set to $\{0.2, 0.4, 0.6, 0.8, 1.0\}$; $\tau$ is the temperature parameter and $q$ is the assessment score of the given image.

### 4.2 PROMPT ENSEMBLE STRATEGY

We introduce the prompt ensemble strategy, which incorporates more prompt groups to derive the final assessment score, thereby achieving a more comprehensive understanding of image quality and aesthetics. For instance, we can use *e.g.*, {`extremely blurry`, `blurry`, `fair`, `sharp`, `extremely sharp`} as another five text levels. Now, the final assessment score $q_f$ is the average of all prompt groups and it can be described as:

$$q_f = \frac{\sum_{i=1}^{m} q_i}{m}, \tag{8}$$

where $m$ denotes the number of prompt groups. This strategy can more fully utilize the multi-modal understanding capabilities of the pre-trained UniQA and demonstrates non-negligible performance improvements in zero-shot (Table 4) and few-label supervised learning (Table 5). The details of ensemble prompts are attached in supplementary material.

Table 1: Results on IQA datasets. **Black** and **blue** numbers in bold represent the best and second best, respectively. Higher SRCC and PLCC imply better performance.

| Method | TID2013 SRCC | TID2013 PLCC | CSIQ SRCC | CSIQ PLCC | KADID SRCC | KADID PLCC | CLIVE SRCC | CLIVE PLCC | KonIQ SRCC | KonIQ PLCC | SPAQ SRCC | SPAQ PLCC |
|---|---|---|---|---|---|---|---|---|---|---|---|---|
| WaDIQaM (Bosse et al., 2017) | 0.835 | 0.855 | 0.852 | 0.844 | 0.739 | 0.752 | 0.682 | 0.671 | 0.804 | 0.807 | 0.840 | 0.845 |
| DBCNN (Zhang et al., 2018) | 0.816 | 0.865 | 0.946 | 0.959 | 0.851 | 0.856 | 0.851 | 0.869 | 0.875 | 0.884 | 0.911 | 0.915 |
| MetaIQA (Zhu et al., 2020) | 0.856 | 0.868 | 0.899 | 0.908 | 0.762 | 0.775 | 0.802 | 0.835 | 0.850 | 0.887 | - | - |
| PaQ-2-PiQ (Ying et al., 2020) | 0.862 | 0.856 | 0.899 | 0.902 | 0.840 | 0.849 | 0.844 | 0.842 | 0.872 | 0.885 | - | - |
| HyperIQA (Su et al., 2020) | 0.840 | 0.858 | 0.923 | 0.942 | 0.852 | 0.845 | 0.859 | 0.882 | 0.906 | 0.917 | 0.911 | 0.915 |
| TReS (Golestaneh et al., 2022) | 0.863 | 0.883 | 0.922 | 0.942 | 0.859 | 0.858 | 0.846 | 0.877 | 0.915 | 0.928 | - | - |
| MUSIQ (Ke et al., 2021) | 0.773 | 0.815 | 0.871 | 0.893 | 0.875 | 0.872 | 0.702 | 0.746 | 0.916 | 0.928 | 0.918 | 0.921 |
| DACNN (Pan et al., 2022) | 0.871 | 0.889 | 0.943 | 0.957 | 0.905 | 0.905 | 0.866 | 0.884 | 0.901 | 0.912 | 0.915 | 0.921 |
| DEIQT (Qin et al., 2023) | **0.892** | **0.908** | 0.946 | **0.963** | 0.889 | 0.887 | 0.875 | 0.894 | 0.921 | 0.934 | 0.919 | 0.923 |
| LIQE (Zhang et al., 2023b) | - | | 0.936 | 0.939 | 0.930 | 0.931 | **0.904** | **0.911** | 0.919 | 0.908 | - | - |
| Re-IQA (Saha et al., 2023) | 0.804 | 0.861 | **0.947** | 0.960 | 0.872 | 0.885 | 0.840 | 0.854 | 0.914 | 0.923 | 0.918 | 0.925 |
| LoDA (Xu et al., 2024) | 0.869 | 0.901 | - | - | **0.931** | **0.936** | 0.876 | 0.899 | 0.932 | **0.944** | **0.925** | **0.928** |
| Q-Align (Wu et al., 2023b) | - | - | 0.915 | 0.936 | 0.869 | 0.927 | **0.931** | 0.921 | **0.935** | 0.934 | - | - |
| Ours | **0.916** | **0.931** | **0.963** | **0.973** | **0.940** | **0.943** | 0.890 | 0.905 | **0.933** | **0.941** | **0.924** | **0.928** |

Table 2: Results on AVA.

| Method | SRCC | PLCC |
|---|---|---|
| NIMA | 0.612 | 0.636 |
| MaxViT | 0.708 | 0.745 |
| APM | 0.709 | - |
| MUSIQ | 0.726 | 0.738 |
| MLSP | 0.756 | 0.757 |
| TANet | 0.758 | 0.765 |
| MILNet | 0.732 | 0.753 |
| EAT | 0.759 | 0.77 |
| VILA | **0.774** | **0.774** |
| Ours | **0.776** | **0.776** |

Table 3: Results on AADB dataset.

| Method | SRCC | PLCC |
|---|---|---|
| NIMA | 0.708 | 0.711 |
| MLSP | 0.725 | 0.726 |
| MUSIQ | 0.706 | 0.712 |
| PA-IAA | 0.720 | 0.728 |
| HIAA | 0.739 | - |
| TANet | 0.738 | 0.737 |
| Celona *et al.* | 0.757 | 0.762 |
| TAVAR | **0.761** | **0.763** |
| Ours | **0.786** | **0.787** |

Table 4: SRCC on the zero-shot setting. * denotes using ensemble prompts. The results of other methods are pre-trained on FLIVE.

| Method | CLIVE | KonIQ | AGIQA-3K |
|---|---|---|---|
| DBCNN | 0.724 | 0.716 | 0.645 |
| PaQ-2-PiQ | 0.738 | 0.755 | 0.502 |
| HyperIQA | 0.735 | 0.758 | 0.629 |
| TReS | 0.740 | 0.713 | 0.646 |
| DEIQT | **0.781** | **0.733** | - |
| CLIP* | 0.746 | 0.592 | 0.646 |
| Ours | 0.638 | 0.667 | **0.744** |
| Ours* | **0.790** | **0.806** | **0.752** |

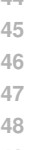
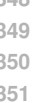
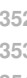
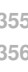
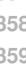

# 5 EXPERIMENTS

## 5.1 DATASETS

We employ the IQA dataset FLIVE (Ying et al., 2020) and the IAA dataset AVA (Murray et al., 2012) for quality- and aesthetics-related captioning, respectively, and AVA-Captions (Ghosal et al., 2019) to provide authentic aesthetic comments. We evaluate the performance on typical IQA and IAA datasets, including seven IQA datasets and two IAA datasets.

**IQA Dataset**. For the IQA task, four synthetic datasets, including LIVE (Sheikh et al., 2006), CSIQ (Larson & Chandler, 2010), TID2013 (Ponomarenko et al., 2013), KADID (Lin et al., 2019), and three authentic datasets of CLIVE (Ghadiyaram & Bovik, 2015), KonIQ (Hosu et al., 2020), SPAQ (Fang et al., 2020), are used for performance evaluation. FLIVE (Ying et al., 2020) is an authentic IQA dataset that contains 39,810 images. We employ an AIGC-generated IQA dataset, AGIQA-3K (Li et al., 2023b), to evaluate the generalization capability of our UniQA. Details of the datasets can be found in appendix.

**IAA Dataset**. For the IAA task, we conduct experiments on AVA (Murray et al., 2012) and AADB (Kong et al., 2016) datasets. AVA comprises 250k images, with the test set of 19,928 images. AADB dataset consists of 10,000 images in total, with 8,500 images for training, 500 images for validation, and 1,000 images for testing.

**AVA-Captions Dataset**. AVA-Captions (Ghosal et al., 2019) offer multiple human-annotated comments for each AVA image. To avoid potential data leakage, we strictly follow the official data split of AVA, results in a pre-training image-text dataset comprising 234,090 images paired with 3.0 million captions.

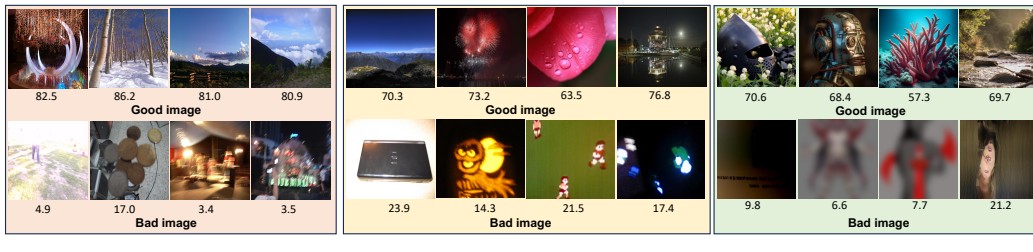

Figure 4: The image retrieval results on three dataset with varied prompts. The number below the image is its MOS label. Zoom in for a better view.

Table 5: SRCC results using few labels for training. $^*$ denotes using ensemble prompts.

| Method | CLIVE | | | KonIQ | | | LIVE | | |
|---|---|---|---|---|---|---|---|---|---|
| | 50 | 100 | 200 | 50 | 100 | 200 | 50 | 100 | 200 |
| HyperIQA (Su et al., 2020) | 0.648 | 0.725 | 0.790 | 0.615 | 0.710 | 0.776 | 0.892 | 0.912 | 0.929 |
| TReS (Golestaneh et al., 2022) | 0.670 | 0.751 | 0.799 | 0.713 | 0.719 | 0.791 | 0.901 | 0.927 | **0.957** |
| ResNet50 (He et al., 2016) | 0.576 | 0.611 | 0.636 | 0.635 | 0.670 | 0.707 | 0.871 | 0.906 | 0.922 |
| CLIP (Radford et al., 2021) | 0.664 | 0.721 | 0.733 | 0.736 | 0.770 | 0.782 | 0.896 | 0.923 | 0.941 |
| CONTRIQUE (Madhusudana et al., 2022) | 0.695 | 0.729 | 0.761 | 0.733 | 0.794 | 0.821 | 0.891 | 0.922 | 0.943 |
| CLIPIQA (Wang et al., 2023) | 0.646 | 0.611 | 0.642 | 0.579 | 0.620 | 0.667 | 0.633 | 0.724 | 0.784 |
| Re-IQA (Saha et al., 2023) | 0.591 | 0.621 | 0.701 | 0.685 | 0.723 | 0.754 | 0.884 | 0.894 | 0.929 |
| DEIQT (Qin et al., 2023) | 0.667 | 0.718 | 0.812 | 0.638 | 0.682 | 0.754 | 0.920 | **0.942** | 0.955 |
| GRepQ (Srinath et al., 2024) | **0.760** | **0.791** | **0.822** | **0.812** | **0.832** | **0.855** | **0.926** | 0.937 | 0.953 |
| Ours | 0.813 | 0.836 | 0.850 | 0.772 | 0.842 | 0.870 | 0.962 | 0.956 | 0.974 |
| Ours$^*$ | **0.828** | **0.849** | **0.853** | **0.844** | **0.860** | **0.876** | **0.963** | **0.958** | **0.976** |

## 5.2 IMPLEMENTATION DETAILS

We use CLIP-B/16 (Radford et al., 2021) as our VLM for pre-training and LLaVA-1.5-7B (Liu et al., 2023b;a) as our MLLM for captioning. We pre-train the model using Adam optimizer (Kingma & Ba, 2014) with a learning rate of 5e-6 and weight decay of 0.2. The model is trained for 5 epochs with a batch size of 960. We set $K = 4$ to refine the AVA-Captions dataset. We use MSE loss to optimize the adapter on downstream tasks and different training settings according to the task and size of datasets. More training details are provided in the appendix. For each IQA dataset, 80% of the images are used for training and the remaining 20% for testing. We repeat this process 10 times to mitigate the performance bias and the medians of SRCC and PLCC are reported. For the IAA datasets, we follow the standard data splits.

## 5.3 MAIN RESULTS

**Results on IQA task.** Table 1 reports the performance of the SOTA IQA methods on six typical IQA datasets. The results of LIVE (Sheikh et al., 2006) are presented in the supplementary material due to page limitations. Our method demonstrates a substantial superiority over existing SOTA models across a diverse range of datasets, fully confirming the effectiveness and excellence of our method in precisely characterizing image quality.

**Results on IAA task.** We report the experimental results on the AVA (Murray et al., 2012) and AADB (Kong et al., 2016) datasets in Table 2 and Table 3, respectively. Given that the pre-trained model acquired a unified and robust image assessment perception, it can also achieve SOTA results after fine-tuning on these two datasets. These results validate that our method can be effectively applied to both IQA and IAA domains.

## 5.4 GENERALIZATION CAPABILITY VALIDATION

Table 4 evaluate the generalization capability of our model. Unlike previous methods that train on one dataset and test on others, we directly utilize the pre-trained UniQA and textual prompts for image quality assessment. This presents a more challenging setting as the model isn't optimized on MOS labels. As observed, our method achieves the best performance on these three datasets. Notably, our method demonstrates excellent performance on AIGC-generated images AGIQA-3K (Li

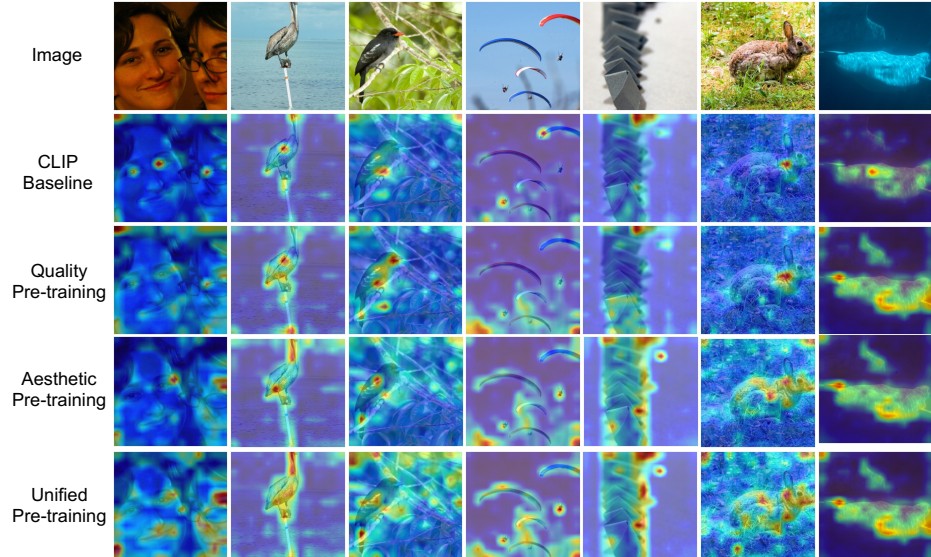

Figure 5: Grad-CAM (Selvaraju et al., 2017) visualization of different pre-training for prompt "blurry image". Through pre-training, the model focuses more on noisy objects and backgrounds.

et al., 2023b), which are markedly different from images of natural scenes. These results demonstrate the strong generalization capability of our UniQA. Additionally, the UniQA outperforms the original CLIP significantly, proving the effectiveness of our quality- and aesthetics-related pre-training.

We use different text queries to calculate the image-text similarity and rank them to achieve zero-shot image retrieval. Figure 4 demonstrates the visualization of the top retrieval results. We notice that the retrieved results of "good image" exhibit sharp and aesthetically pleasing images, whereas "bad image" prompts retrieve blurry, poor lighting and meaningless images. These examples provide qualitative evidence of the quality and aesthetic knowledge captured by the pre-trained model.

### 5.5 DATA-EFFICIENT LEARNING

The pre-trained model acquires extensive image assessment knowledge, providing robust priors for downstream tasks. Consequently, our model can deliver impressive performance with limited data. To validate this, we randomly select subsets of 50, 100, and 200 samples from the training set for training and evaluate them on the same test data as full-data supervised learning. We report the median performance across 10 times in Table 5. Our method notably outperforms the second-best model GRepQ by a substantial margin, even though GRepQ is specifically designed for data-efficient learning. These results thoroughly demonstrate the potent capability of our method to learn image quality even when only a few labels are available. Additionally, several insightful observations can be drawn from Table 5. Firstly, the prompt ensemble strategy markedly enhances model performance in the data-efficient setting. This is attributed to its ability to more fully leverage the extensive knowledge of the pre-

Table 6: Ablation on IQA (CLIVE and KonIQ) and IAA (AVA) datasets with SRCC metrics.

| $Y_{IQA}$ | $Y_{IAA}$ | $Y_{IAA}^{+}$ | CLIVE | KonIQ | AVA |
|---|---|---|---|---|---|
| Ablation on different pre-training data | | | | | |
| × | × | × | 0.865 | 0.907 | 0.748 |
| ✓ | × | × | 0.871 | 0.914 | 0.755 |
| × | ✓ | × | 0.871 | 0.917 | 0.755 |
| ✓ | ✓ | × | 0.874 | 0.918 | 0.756 |
| × | × | ✓ | 0.875 | 0.928 | 0.773 |
| × | ✓ | ✓ | 0.877 | 0.930 | 0.774 |
| ✓ | ✓ | ✓ | **0.890** | **0.933** | **0.776** |
| Ablation on data purification strategy | | | | | |
| w/o Strategy | | | 0.876 | 0.929 | 0.772 |
| IR Strategy | | | 0.879 | 0.931 | 0.774 |
| AR Strategy | | | 0.885 | 0.930 | 0.774 |
| AIR Strategy | | | **0.890** | **0.933** | **0.776** |
| Ablation on the proposed adapter | | | | | |
| Single Prompt | | | 0.705 | 0.920 | 0.765 |
| Antonym Prompt | | | 0.875 | 0.928 | 0.771 |
| Ours adapter | | | **0.890** | **0.933** | **0.776** |
| Ablation on different MLLMs | | | | | |
| LLaVA-v1.5-7B | | | 0.871 | 0.914 | 0.755 |
| LLaVA-v1.5-13B | | | 0.872 | 0.914 | 0.757 |
| Sphinx | | | 0.874 | 0.916 | 0.758 |
| QWen-VL | | | 0.870 | 0.913 | 0.757 |
| LLaVa-7B+QWen | | | 0.875 | 0.916 | 0.758 |
| Sphinx+QWen | | | **0.877** | **0.918** | **0.759** |

trained model. Secondly, the impact of prompt ensemble is slight on synthetic datasets. This is likely due to the limited image variety within synthetic datasets, making a single prompt sufficient for such scenarios.

## 5.6 ABLATION STUDIES

**Impact of different pre-training data.** Table 6 shows the effect of different pre-training data. We observe that unified pre-training achieves the optimal performance on both tasks. In addition, we derive some meaningful observations. (1) Using either the generated $Y_{IQA}$ or $Y_{IAA}$ improves the performance of both IQA and IAA tasks, proving the mutual benefit of these two tasks and the effectiveness of MLLMs captioning. (2) Unifying $Y_{IQA}$ and $Y_{IAA}$ datasets does not lead to significant improvements. We believe this is because the MLLMs-generated text tends to have similar sentence structures (Liu et al., 2023c) and perception representation, limiting the diversity provided for multimodal learning. (3) Pre-training with refined authentic $Y_{IAA}^+$ shows significant improvement on two tasks, reflecting that human-annotated comments can provide a more comprehensive and effective representation for the model.

Figure 5 illustrates the Grad-CAM (Selvaraju et al., 2017) visualization of different pre-training. We can notice that after quality-related and aesthetic pre-training, the model pays more attention to blurred subjects and noisy backgrounds. This effect becomes more pronounced with unified pre-training, underscoring the advantages of such a unified approach. In addition, the unified pre-training can focus on the areas of quality-related and aesthetic pre-training at the same time. This shows that unified training can learn common representations of the two tasks.

**Effectiveness of data purification strategy.** The second part of Table 6 illustrates the ablation study of the data purification strategy. It can be observed that employing either AR or IR strategy to purify data can improve the model's performance of both IQA and IAA tasks. These results validate the benefit of obtaining aesthetically relevant and semantically rich textual descriptions for the model. Finally, when combining these two strategies, it achieves the best performance.

**Effectiveness of the Multi-Cue Integration Adapter.** The third part of Table 6 shows the ablation study of the proposed adapter. "Single Prompt" denotes using the similarity between the text "good image" and images as the assessment score directly, while "Antonym Prompt" represents using the relative weights of texts "good image" and "bad image" to weight the predefined score. It is evident that the "Single Prompt" is considerably inferior to the "Antonym Prompt", showing the limitations of using semantic similarity as score directly. Our method integrates more cues into the "Antonym Prompt" to comprehensively assess images, thereby achieving optimal performance.

**Ablation on different MLLMs.** The bottom part of Table 6 presents the ablation study of various MLLMs. We generate $Y_{IQA}$ via different MLLMs for pre-training. It is evident that using different MLLMs exhibits similar performance, while ensembling different MLLMs can boost performance. This indicates that MLLMs are capable of generating accurate captions with our text-guided prompt, and enhancing caption diversity can further improve performance. Considering resource limitations, we use LLaVa-7B and will integrate more MLLMs in the future.

## 6 CONCLUSION AND DISCUSSION

This paper introduces UniQA, which leverages unified vision-language pre-training to address quality and aesthetic assessment problems concurrently. We construct a high-quality image-text dataset about quality and aesthetics with the assistance of MLLMs. Through large-scale pre-training on this dataset, UniQA learns shared and effective representations of IQA and IAA tasks, benefiting both tasks. Additionally, we propose a Multi-Cue Integration Adapter to effectively adapt the pre-trained UniQA to downstream assessment tasks. Our method achieves state-of-the-art performance on both IQA and IAA tasks, and demonstrates powerful zero-shot and few-label image assessment capabilities.

**Limitations and future work.** MLLMs often generate captions with similar sentence structures and semantic expressions, restricting their ability to provide diverse and enriched representations for multimodal learning. Future work will explore other techniques to address this issue, including integrating various MLLMs for captioning and employing in-context learning methods.

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

# A  MORE DISCUSSION AND DETAILS

## A.1  DISCUSSION ABOUT THE AIR

We propose the Aesthetics-relevance and Informativeness Rank (AIR) to select the high-quality texts corresponding to an image. The AIR can be expressed as follows:

$$\text{AIR} = \text{Rank}((\text{AR}^1 + \text{IR}^1), \cdots, (\text{AR}^n + \text{IR}^n)). \tag{9}$$

where $\text{AR}$ and $\text{IR}$ denote the Aesthetics-relevance Rank and Informativeness Rank, respectively; $n$ is the number of comments corresponding to an image. For simplicity, we directly take the summation of IR and AR to reflect the semantic relevance and richness of the text. In fact, we can introduce two factors ($\alpha$ and $\beta$) to purify the data more flexibly. Now, the modified $\text{AIR}_\text{m}$ can be formulated as:

$$\text{AIR}_\text{m} = \text{Rank}((\alpha\text{AR}^1 + \beta\text{IR}^1), \cdots, (\alpha\text{AR}^n + \beta\text{IR}^n)). \tag{10}$$

For instance, we can use large $\alpha$ for the highly noisy data. With this strategy, we can more flexibly purify data based on data quality.

Table 7: Details of different IQA datasets.

| Dataset | Dataset Type | Dataset Size | Number of distortions |
|---|---|---|---|
| LIVE Sheikh et al. (2006) | Synthetic | 799 | 5 |
| CSIQ Larson & Chandler (2010) | Synthetic | 866 | 5 |
| TID2013 Ponomarenko et al. (2013) | Synthetic | 3,000 | 24 |
| KADID Lin et al. (2019) | Synthetic | 10,125 | 25 |
| CLIVE Ghadiyaram & Bovik (2015) | Authentic | 1,162 | - |
| KonIQ Hosu et al. (2020) | Authentic | 10,073 | - |
| SPAQ Fang et al. (2020) | Authentic | 11,000 | - |
| FLIVE Ying et al. (2020) | Authentic | 39,810 | - |
| AGIQA-3K Li et al. (2023b) | Authentic | 2,982 | - |

Table 8: Results on LIVE dataset Sheikh et al. (2006). **Black** and **blue** numbers in bold represent the best and second best, respectively. Higher SRCC and PLCC imply better performance.

| | LIVE | |
|---|---|---|
| Method | SRCC | PLCC |
| DIIVINE Bosse et al. (2017) | 0.892 | 0.908 |
| BRISQUE Mittal et al. (2012a) | 0.929 | 0.944 |
| ILNIQE Zhang et al. (2015) | 0.902 | 0.906 |
| BIECON Kim & Lee (2016) | 0.958 | 0.961 |
| MEON Ma et al. (2017) | 0.951 | 0.955 |
| WaDIQaM Bosse et al. (2017) | 0.960 | 0.955 |
| DBCNN Zhang et al. (2018) | 0.968 | 0.971 |
| MetaIQA Zhu et al. (2020) | 0.960 | 0.959 |
| PaQ-2-PiQ Ying et al. (2020) | 0.959 | 0.958 |
| HyperIQA Su et al. (2020) | 0.962 | 0.966 |
| TReS Golestaneh et al. (2022) | 0.969 | 0.968 |
| MUSIQ Ke et al. (2021) | 0.940 | 0.911 |
| DACNN Pan et al. (2022) | 0.978 | 0.980 |
| DEIQT Qin et al. (2023) | **0.980** | **0.982** |
| LIQE Zhang et al. (2023b) | 0.970 | 0.951 |
| Re-IQA (Saha et al., 2023) | 0.970 | 0.971 |
| LoDA (Xu et al., 2024) | 0.975 | 0.979 |
| Ours | **0.981** | **0.983** |

## A.2  DETAILS OF DATASETS AND EVALUATION CRITERIA

We list the details of the datasets used in our work in Table 7, including the dataset type, dataset size and number of distortion types. Since the distortions of authentic datasets are diverse, their number cannot be counted.

We employ Spearman's Rank-order Correlation Coefficient (SRCC) and Pearson's Linear Correlation Coefficient (PLCC) as criteria to measure the performance of IQA and IAA models. They reflect the prediction monotonicity and prediction accuracy of the model, respectively. Both SRCC and PLCC range from 0 to 1. Higher values of SRCC and PLCC indicate better performance.

Table 9: Training settings for different datasets.

| Dataset | Task | Epoch | Batch size | Learning rate |
|---|---|---|---|---|
| LIVE Sheikh et al. (2006) | IQA | 50 | 8 | 2e-4 |
| CSIQ Larson & Chandler (2010) | IQA | 50 | 8 | 2e-4 |
| TID2013 Ponomarenko et al. (2013) | IQA | 20 | 8 | 2e-4 |
| KADID Lin et al. (2019) | IQA | 20 | 8 | 2e-4 |
| CLIVE Ghadiyaram & Bovik (2015) | IQA | 50 | 8 | 2e-4 |
| KonIQ Hosu et al. (2020) | IQA | 20 | 8 | 2e-4 |
| SPAQ Fang et al. (2020) | IQA | 20 | 8 | 2e-4 |
| AVA Murray et al. (2012) | IAA | 20 | 128 | 5e-4 |
| AADB Kong et al. (2016) | IAA | 20 | 8 | 5e-4 |

### A.3 MORE IMPLEMENTATION DETAILS

For the pre-training, we employ the same training strategy as CLIP Radford et al. (2021) to pre-train our UniQA. The pre-training is resource-friendly and takes *less than an hour* at a time. When fine-tuning the adapter for downstream assessment tasks, we use different training settings according to the task and size of the dataset. Table 9 shows the detailed training setting for the different datasets. We follow the typical training strategy to fine-tune each dataset, including random cropping and random horizontal flipping. Since different datasets have different MOS scales, we scale their range to [0, 1] through normalization. During inference, we typically crop an input image into 10 image patches and take their average as the quality score of this image Su et al. (2020); Qin et al. (2023). We use the resolution of $224 \times 224$ for training and testing. All experiments are conducted on two A100 GPUs.

### A.4 PROMPT ENSEMBLE

When applying our UniQA to zero-shot and few-label settings, prompt ensemble is a useful strategy to improve performance. Table 10 shows the prompt groups used in these two settings. Note that the prompts used in AGIQA-3K are different from other IQA datasets. This is because distortions in AIGC-generated images and authentic images tend to be different. For example, distortions in authentic images may come from camera shake. However, distortions in AIGC-generated images typically come from low-quality content, such as meaningless content and distorted poses. Therefore, we use "content" to prompt the pre-trained multimodal model for the AGIQA-3K dataset.

## B MORE EXPERIMENTAL RESULTS

### B.1 COMPARISON RESULTS ON LIVE

Tab .8 shows the comparison results with other methods on LIVE dataset Sheikh et al. (2006). We can observe that our method also achieves state-of-the-art (SOTA) performance, verifying the effectiveness of our method.

### B.2 MORE GENERALIZATION EXPERIMENTS

In this section, we conduct more experiments to further verify the generalization capability of UniQA. We evaluate our model on three dataset, including AIGC IQA datatset AIGIQA-20K, the enhanced colonoscopy image quality assessment dataset (ECIQAD) and the AI-Generated Image Naturalness (AGIN) dataset.

**AIGIQA-20K**. AIGIQA-20K Li et al. (2024) is a large-scale AI-generated image quality assessment dataset. It consists of 20,000 images in total, with 14,000 images for training, 2,000 images for

Table 10: Text prompts used in zero-shot and few-label learning.

| Task | Prompt |
|---|---|
| CLIVE, KonIQ, LIVE | {bad, poor, fair, good, perfect} with image |
| | {extremely blurry, blurry, fair, sharp, extremely sharp} with image |
| | {extremely noisy, noisy, fair, noise-free, extremely noise-free} with image |
| | {extremely low-quality, low-quality, fair, high-quality, extremely high-quality} with image |
| AGIQA-3K | {bad, poor, fair, good, perfect} with image {bad, poor, fair, good, perfect} with content |

Table 11: Results on AIGIQA-20K dataset Li et al. (2024). * indicates that we also unfreeze the backbone for training with a smaller learning rate of 2e-6.

| Method | SRCC | PLCC |
|---|---|---|
| CLIPIQA | 0.331 | 0.483 |
| CLIIQA+Finetune | 0.786 | 0.712 |
| CNNIQA | 0.330 | 0.367 |
| CNNIQA+Finetune | 0.597 | 0.591 |
| Q-Align | 0.746 | 0.742 |
| DBCNN Zhang et al. (2018) | 0.471 | 0.512 |
| DBCNN+Finetune | **0.851** | 0.869 |
| Ours | 0.576 | 0.563 |
| Ours+Finetune | 0.830 | **0.885** |
| Ours+Finetune* | **0.858** | **0.901** |

validation, and 4,000 images for testing. We test our model on the zero-shot and fine-tuning setting. For fine-tuning, we use a learning rate of 2e-4 for the adapter and train the model for 10 epochs. As shown in Table 11, we can notice that our model achieves competitive results in both settings. These results verify the excellent generalization ability of UniQA on AI-generated images.

**ECIQAD.** ECIQAD (Yue et al., 2023) is an enhanced colonoscopy image quality assessment dataset containing 2400 images in total. We repeat the experiment 10 times with an 8:2 data split and report the median results. We train the model for 50 epochs. Other training settings are the same as AIGIQA-20K. The experimental results are shown in Table 12. Our method achieves SOTA results on ECIQAD. Since the ECIQAD dataset is quite different from natural images, these results fully demonstrate the strong generalization and image assessment capabilities of our method.

**AGIN.** AGIN (Chen et al., 2023) is an AI-Generated Image Naturalness (AGIN) dataset, which includes 6049 images. We randomly split the training, validation, and testing set into 7:1:2. We repeat this process 5 times and report the average performance as the final experimental results. We train the model for 20 epochs. As shown in Table 13, although our approach is not specifically designed for the AI naturalness problem, our method achieves competitive results compared to specific designed JOINT (Chen et al., 2023) and other methods. These results further demonstrate the generalization ability of our model.

## B.3 PLCC COMPARISON IN THE DATA-EFFICIENT SETTING

The Pearson's Linear Correlation Coefficient (PLCC) comparisons for our method against other IQA methods corresponding to the table in the main paper are provided in Tab .14. We note that our method outperforms all other methods in terms of PLCC metric.

Table 12: Results on ECIQAD (Yue et al., 2023). * indicates that we also unfreeze the backbone for training with a smaller learning rate of 2e-6.

| Method | SRCC | PLCC |
|---|---|---|
| BRISQUE (Mittal et al., 2012a) | 0.436 | 0.459 |
| BIQME (Gu et al., 2017) | 0.770 | 0.768 |
| BPRI (Min et al., 2017) | 0.152 | 0.181 |
| FRIQUEE (Ghadiyaram & Bovik, 2017) | 0.663 | 0.656 |
| CIQA (Chen et al., 2021) | 0.738 | 0.735 |
| ECIQ (Ke et al., 2021) | 0.839 | 0.842 |
| Ours | **0.873** | **0.887** |
| Ours* | **0.918** | **0.928** |

Table 13: Results on AGIN (Chen et al., 2023). * indicates that we also unfreeze the backbone for training with a smaller learning rate of 2e-6.

| Methods | Technical | | Rationality | | Naturalness | |
|---|---|---|---|---|---|---|
| | SRCC | PLCC | SRCC | PLCC | SRCC | PLCC |
| BRISQUE (Mittal et al., 2012a) | 0.4867 | 0.4909 | 0.3608 | 0.3684 | 0.3745 | 0.4067 |
| NIQE (Mittal et al., 2012b) | 0.4235 | 0.4279 | 0.3144 | 0.3211 | 0.3358 | 0.3378 |
| DBCNN (Zhang et al., 2018) | 0.7623 | 0.7661 | 0.6834 | 0.6838 | 0.7057 | 0.7128 |
| HyperIQA (Su et al., 2020) | 0.7752 | 0.7806 | 0.7196 | 0.7292 | 0.7365 | 0.7509 |
| MUSIQ (Ke et al., 2021) | 0.7286 | 0.7355 | 0.6974 | 0.7013 | 0.7066 | 0.7103 |
| UNIQUE (Zhang et al., 2021a) | 0.7358 | 0.7434 | 0.6583 | 0.6685 | 0.6772 | 0.6789 |
| MANIQA (Yang et al., 2022) | 0.7763 | 0.7817 | 0.7192 | 0.7217 | 0.7385 | 0.7343 |
| PAIAA (Li et al., 2020) | 0.4763 | 0.4833 | 0.4532 | 0.4596 | 0.4483 | 0.4528 |
| TANet (He et al., 2022) | 0.5367 | 0.5587 | 0.4731 | 0.4762 | 0.4782 | 0.4535 |
| Del. Transf. (He et al., 2023a) | 0.5882 | 0.6134 | 0.5037 | 0.4942 | 0.4805 | 0.4961 |
| SAAN (Yi et al., 2023) | 0.4299 | 0.4380 | 0.4009 | 0.4160 | 0.4196 | 0.4184 |
| JOINT (Chen et al., 2023) | **0.8173** | **0.8235** | 0.7564 | 0.7711 | 0.7986 | 0.8028 |
| JOINT++ (Chen et al., 2023) | **0.8351** | **0.8429** | **0.8033** | **0.8127** | **0.8264** | **0.8362** |
| Ours | 0.7524 | 0.8007 | 0.7728 | 0.7793 | 0.7882 | 0.7979 |
| Ours | 0.7785 | 0.8104 | **0.7898** | **0.7952** | **0.8069** | **0.8171** |

### B.4 MORE COMPARISON RESULTS ON IQA DATASETS

To demonstrate the superiority of our method more comprehensively, we present more comparison results on the typical IQA datasets in Table 15.

### B.5 MORE RESULTS OF ABLATION STUDY

Table 16 shows the SRCC and PLCC results of the ablation study.

## C ANALYSIS OF CONSTRUCTED MULTIMODAL DATASET

In this section, we analyze the constructed multimodal image and text dataset. Our proposed dataset has 273,897 images, with 1,240,915 captions. Next we conduct a detailed analysis of the dataset:

1. Firstly, we compare the data volume of the IQA and IAA datasets in Figure 6. IQA includes 39,807 images and IAA includes 234,090 images. We generate three captions for each IQA image and one caption for each IAA image, resulting 119,421 generated IQA captions and 234,090 IAA captions. From the ablation experiment in Table 6, we notice that $Y_{IQA}$ and $Y_{IAA}$ have similar performance improvements on the model, although $Y_{IAA}$ has more data. Therefore, this shows that the text generated by MLLM tends to have the same structure and lacks diversity, limiting the further improvement of model performance.

2. Secondly, we analyze the number of words in the generated text, as shown in Figure 7. We can see that the number of words in most texts is between 20 and 40. In addition, there are

Table 14: PLCC performance comparison of our method with other NR-IQA methods trained using few labels. * denotes using ensemble prompts.

| Method | LIVEC 50 | 100 | 200 | KonIQ 50 | 100 | 200 | LIVE 50 | 100 | 200 |
|---|---|---|---|---|---|---|---|---|---|
| HyperIQA (Su et al., 2020) | 0.689 | 0.755 | 0.806 | 0.650 | 0.758 | 0.807 | 0.903 | 0.922 | 0.931 |
| TReS (Golestaneh et al., 2022) | 0.702 | 0.776 | 0.813 | 0.740 | 0.748 | 0.824 | 0.916 | 0.948 | 0.960 |
| ResNet50 (He et al., 2016) | 0.580 | 0.629 | 0.660 | 0.661 | 0.693 | 0.716 | 0.872 | 0.908 | 0.920 |
| CLIP (Radford et al., 2021) | 0.676 | 0.739 | 0.758 | 0.749 | 0.790 | 0.802 | 0.891 | 0.924 | 0.942 |
| CONTRIQUE (Madhusudana et al., 2022) | 0.693 | 0.736 | 0.777 | 0.743 | 0.801 | 0.832 | 0.892 | 0.922 | 0.944 |
| CLIPIQA (Wang et al., 2023) | 0.633 | 0.606 | 0.639 | 0.586 | 0.616 | 0.681 | 0.613 | 0.706 | 0.752 |
| Re-IQA (Saha et al., 2023) | 0.620 | 0.650 | 0.701 | 0.689 | 0.693 | 0.757 | 0.876 | 0.892 | 0.931 |
| DEIQT (Qin et al., 2023) | 0.695 | 0.739 | 0.818 | 0.670 | 0.707 | 0.778 | 0.916 | 0.942 | 0.957 |
| GRepQ (Srinath et al., 2024) | 0.772 | 0.798 | 0.835 | 0.793 | 0.816 | 0.840 | 0.929 | 0.936 | 0.957 |
| Ours | 0.819 | 0.854 | 0.866 | 0.815 | 0.861 | 0.890 | 0.952 | 0.959 | 0.970 |
| Ours* | **0.826** | **0.847** | **0.869** | **0.857** | **0.883** | **0.893** | **0.963** | **0.962** | **0.973** |

Table 15: Results on IQA datasets. **Black** and **blue** numbers in bold represent the best and second best, respectively. Higher SRCC and PLCC imply better performance.

| Method | TID2013 SRCC | PLCC | CSIQ SRCC | PLCC | KADID SRCC | PLCC | CLIVE SRCC | PLCC | KonIQ SRCC | PLCC | SPAQ SRCC | PLCC |
|---|---|---|---|---|---|---|---|---|---|---|---|---|
| DIIVINE Bosse et al. (2017) | 0.643 | 0.567 | 0.804 | 0.776 | 0.413 | 0.435 | 0.588 | 0.591 | 0.546 | 0.558 | 0.599 | 0.600 |
| BRISQUE Mittal et al. (2012a) | 0.626 | 0.571 | 0.812 | 0.748 | 0.528 | 0.567 | 0.629 | 0.629 | 0.681 | 0.685 | 0.809 | 0.817 |
| ILNIQE Zhang et al. (2015) | 0.521 | 0.648 | 0.822 | 0.865 | 0.534 | 0.558 | 0.508 | 0.508 | 0.523 | 0.537 | 0.712 | 0.713 |
| BIECON Kim & Lee (2016) | 0.717 | 0.762 | 0.815 | 0.823 | 0.623 | 0.648 | 0.613 | 0.613 | 0.651 | 0.654 | - | - |
| MEON Ma et al. (2017) | 0.808 | 0.824 | 0.852 | 0.864 | 0.604 | 0.691 | 0.697 | 0.710 | 0.611 | 0.628 | - | - |
| WaDIQaM Bosse et al. (2017) | 0.835 | 0.855 | 0.852 | 0.844 | 0.739 | 0.752 | 0.682 | 0.671 | 0.804 | 0.807 | 0.840 | 0.845 |
| DBCNN Zhang et al. (2018) | 0.816 | 0.865 | 0.946 | 0.959 | 0.851 | 0.856 | 0.851 | 0.869 | 0.875 | 0.884 | 0.911 | 0.915 |
| MetaIQA Zhu et al. (2020) | 0.856 | 0.868 | 0.899 | 0.908 | 0.762 | 0.775 | 0.802 | 0.835 | 0.850 | 0.887 | - | - |
| PaQ-2-PiQ Ying et al. (2020) | 0.862 | 0.856 | 0.899 | 0.902 | 0.840 | 0.849 | 0.844 | 0.842 | 0.872 | 0.885 | - | - |
| HyperIQA Su et al. (2020) | 0.840 | 0.858 | 0.923 | 0.942 | 0.852 | 0.845 | 0.859 | 0.882 | 0.906 | 0.917 | 0.911 | 0.915 |
| TReS Golestaneh et al. (2022) | 0.863 | 0.883 | 0.922 | 0.942 | 0.859 | 0.858 | 0.846 | 0.877 | 0.915 | 0.928 | - | - |
| MUSIQ Ke et al. (2021) | 0.773 | 0.815 | 0.871 | 0.893 | 0.875 | 0.872 | 0.702 | 0.746 | 0.916 | 0.928 | 0.918 | 0.921 |
| DACNN (Pan et al., 2022) | 0.871 | 0.889 | 0.943 | 0.957 | 0.905 | 0.905 | 0.866 | 0.884 | 0.901 | 0.912 | 0.915 | 0.921 |
| DEIQT (Qin et al., 2023) | 0.892 | 0.908 | 0.946 | 0.963 | 0.889 | 0.887 | 0.875 | 0.894 | 0.921 | 0.934 | 0.919 | 0.923 |
| LIQE (Zhang et al., 2023b) | - | - | 0.936 | 0.939 | 0.930 | 0.931 | 0.904 | 0.911 | 0.919 | 0.908 | - | - |
| Re-IQA (Saha et al., 2023) | 0.804 | 0.861 | 0.947 | 0.960 | 0.872 | 0.885 | 0.840 | 0.854 | 0.914 | 0.923 | 0.918 | 0.925 |
| CIS (Zhong et al.) | - | - | - | - | - | - | 0.828 | 0.847 | 0.881 | 0.918 | - | - |
| LAR-IQA Avanaki et al. (2024) | - | - | - | - | **0.941** | **0.965** | - | - | - | - | - | - |
| LoDA (Xu et al., 2024) | 0.869 | 0.901 | - | - | 0.931 | 0.936 | 0.876 | 0.899 | 0.932 | 0.944 | 0.925 | 0.928 |
| DP-IQA Fu et al. (2024) | - | - | - | - | - | - | 0.893 | 0.913 | 0.942 | 0.951 | 0.923 | 0.926 |
| Q-Align (Wu et al., 2023b) | - | - | 0.915 | 0.936 | 0.869 | 0.927 | 0.931 | 0.921 | 0.935 | 0.934 | - | - |
| Ours | 0.916 | 0.931 | 0.963 | 0.973 | 0.940 | 0.943 | 0.890 | 0.905 | 0.933 | 0.941 | 0.924 | 0.928 |

also many samples with the number of words between 80-120, which are texts generated by MLLM.

3. Finally, we construct a word cloud for the text data, as shown in Figure 8. It can be seen that the most common words in the text dataset are aesthetic and quality-related words, such as "aesthetics", "quality", "composition", "fair", etc. This indicates that the text of the constructed dataset focuses on image assessment.

## D    DETAILS AND DISCUSSION OF MLLMs CAPTIONING

**Details of MLLMs Captioning.** We use different numbers of captions for IQA and IAA tasks. Considering that the IQA data does not have textual descriptions, we generate three captions with different prompts via MLLMs for the IQA datasets. This method can improve the text diversity of IQA image-text data. For the IAA dataset, we generate one caption for each image because IAA datasets have a large amount of authentic text data. Details of the prompts for quality-related captioning are shown in Figure 9.

**Effectiveness of text guidance.** We visualize the MLLMs-generated captions with/without text guidance to evaluate the effectiveness of text guidance. We take the captioning for IQA datasets as examples. As shown in Figure 13, when the quality of image is high, the MLLMs can output correct caption (see example 1). *However, we can observe that the MLLMs will generate wrong captions*

Table 16: Ablation experiments on two IQA datasets (CLIVE and KonIQ) and one IAA dataset (AVA). Different ablations are distinguished by different backgrounds for better viewing.

| Ablation type | | | CLIVE | | KonIQ | | AVA | |
|---|---|---|---|---|---|---|---|---|
| | | | SRCC | PLCC | SRCC | PLCC | SRCC | PLCC |
| Ablation on different pre-training data | | | | | | | | |
| $Y_{IQA}$ | $Y_{IAA}$ | $Y_{IAA}^+$ | | | | | | |
| × | × | × | 0.865 | 0.886 | 0.907 | 0.924 | 0.748 | 0.747 |
| ✓ | × | × | 0.871 | 0.898 | 0.914 | 0.932 | 0.755 | 0.755 |
| × | ✓ | × | 0.871 | 0.895 | 0.917 | 0.932 | 0.755 | 0.756 |
| ✓ | ✓ | × | 0.874 | 0.892 | 0.918 | 0.932 | 0.756 | 0.757 |
| × | × | ✓ | 0.875 | 0.895 | 0.928 | 0.937 | 0.773 | 0.774 |
| × | ✓ | ✓ | 0.877 | 0.895 | 0.930 | 0.939 | 0.774 | 0.774 |
| ✓ | ✓ | ✓ | **0.890** | **0.905** | **0.933** | **0.941** | **0.776** | **0.776** |
| Ablation on data purification strategy | | | | | | | | |
| w/o Strategy | | | 0.876 | 0.899 | 0.929 | 0.940 | 0.772 | 0.771 |
| IR Strategy | | | 0.879 | 0.898 | 0.931 | 0.941 | 0.774 | 0.774 |
| AR Strategy | | | 0.885 | 0.901 | 0.930 | 0.942 | 0.774 | 0.773 |
| AIR Strategy | | | **0.890** | **0.905** | **0.933** | **0.941** | **0.776** | **0.776** |
| Ablation on the proposed adapter | | | | | | | | |
| Single Prompt | | | 0.705 | 0.720 | 0.920 | 0.931 | 0.765 | 0.765 |
| Antonym Prompt | | | 0.875 | 0.897 | 0.928 | 0.938 | 0.771 | 0.772 |
| Ours adapter | | | **0.890** | **0.905** | **0.933** | **0.941** | **0.776** | **0.776** |
| Ablation on different MLLMs | | | | | | | | |
| LLaVA-v1.5-7B | | | 0.871 | 0.898 | 0.914 | 0.932 | 0.755 | 0.755 |
| LLaVA-v1.5-13B | | | 0.872 | 0.897 | 0.914 | 0.929 | 0.757 | 0.759 |
| Sphinx | | | 0.874 | 0.902 | 0.916 | 0.931 | 0.758 | 0.758 |
| QWen-VL | | | 0.870 | 0.895 | 0.913 | 0.930 | 0.757 | 0.758 |
| LLaVa-7B+QWen | | | 0.875 | 0.899 | 0.916 | 0.930 | 0.758 | 0.757 |
| Sphinx+QWen | | | **0.877** | **0.908** | **0.918** | **0.934** | **0.759** | **0.760** |

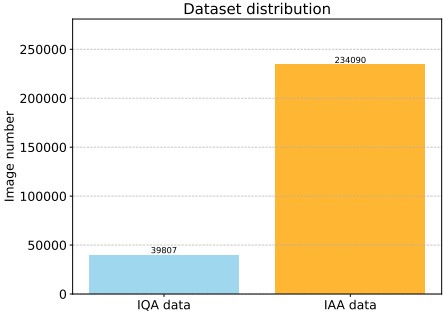

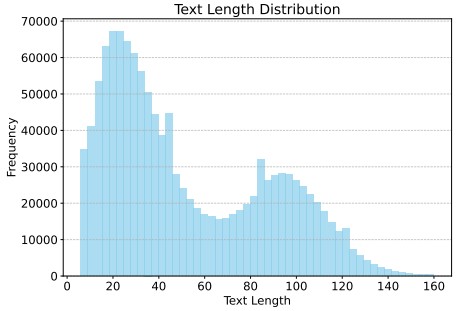

Figure 6: The number of images for the two tasks in the constructed dataset.

Figure 7: The length distribution of texts in the dataset.

*when low-quality images are sent to MLLMs and the text guidance is not provided (see example 2-4 of Figure 13).* As a result, the text guidance is useful and necessary for MLLMs captioning.

# E MORE VISUALIZATION

## E.1 MORE EXAMPLES OF ZERO-SHOT RETRIEVAL

We show more examples of zero-shot retrieval in Figure 10, Figure 11 and Figure 12. We explore the model's retrieval capabilities in terms of color and light. We can observe that the images of "Good color" show harmonious and vivid colors while the images of "Bad color" present messy

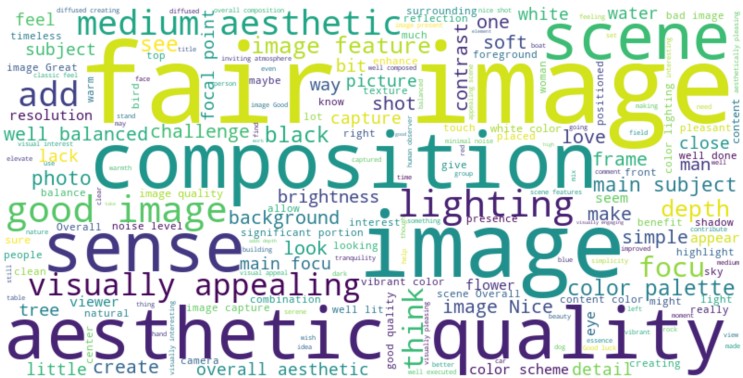

Figure 8: Word cloud of the constructed dataset.

---

**Details of quality-related captioning**

**MOS of image**: 6.71 → **Text level**: good

**Instruction 1**: What you are given is a good quality image. Evaluate image quality based on factors such as *sharpness, color balance, and noise levels*.
**Output 1**: The image is of three hot air balloons flying in the sky. The balloons are of different sizes and are positioned at various heights. The sky is clear and blue, providing a perfect backdrop for the balloons. The resolution of the image is good, allowing for clear visibility of the balloons and their details. Overall, the image quality is excellent, capturing the essence of the hot air balloons in flight.

**Instruction 2**: Giving you a good quality image. Describe the quality of an image based on visual descriptors such as *resolution, noise, brightness*.
**Output 2**: The image is of good quality, with a resolution that allows for clear visibility of the three hot air balloons flying in the sky. The noise level is low, and the brightness of the scene is well-balanced, making it an enjoyable view. Overall, the image is high quality and has clear content.

**Instruction 3**: This is a good quality (*sharpness, contrast, distortion*) image. Assuming you are a human observer, please describe your feelings when you see this image.
**Output 2**: As a human observer, I would feel a sense of wonder and excitement upon seeing this image. The two hot air balloons flying in the sky, each with a unique design, create a visually appealing and captivating scene. The contrast between the vibrant colors of the balloons and the clear blue sky adds to the overall beauty of the image.

Figure 9: Details of the prompts of quality-related captioning. We generate three captions for each IQA image using three different prompts.

colors. For the retrieval on light, the images of "Good light" are clear and well-lit and the images of "Bad light" demonstrate dimly lit environments and poor lighting. These visualization results further demonstrate the strong capability of our model in understanding concepts related to quality and aesthetics.

E.2 VISUALIZATION OF DATA PURIFICATION STRATEGY

In Figure 14, we visualize the comments sorted by the proposed strategies. As observed, the Aesthetics-relevance Rank (AR) can sort the aesthetics-related comments first. Further integrating the Informativeness Rank (IR), we can obtain aesthetically relevant comments with rich semantics.

E.3 MORE EXAMPLES OF TEXT GENERATED BY MLLMS

In Figure 15, we show more examples of the captions generated by MLLMs. We can observe that with our prompt design, MLLMs can output correct fine-grained quality- and aesthetics-related descriptions.

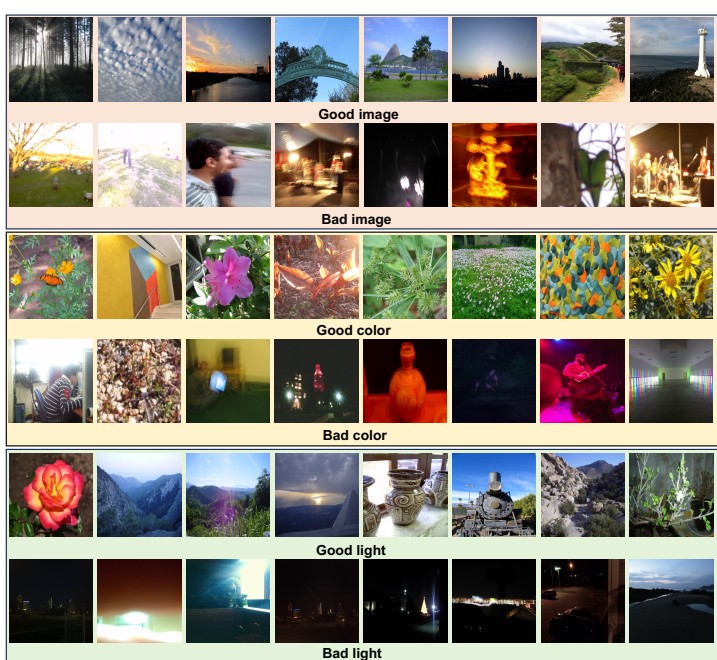

Figure 10: More image retrieval results with various text as queries on CLIVE Ghadiyaram & Bovik (2015).

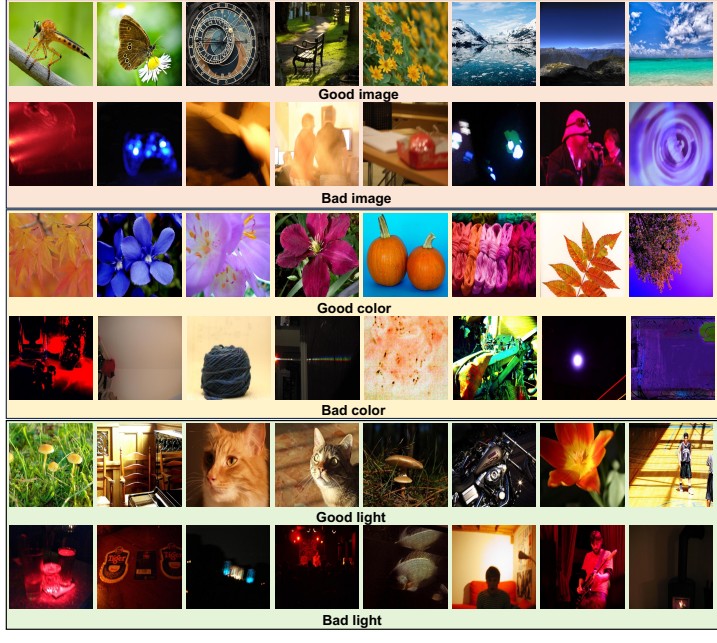

Figure 11: More image retrieval results with various text as queries on KonIQ Hosu et al. (2020).

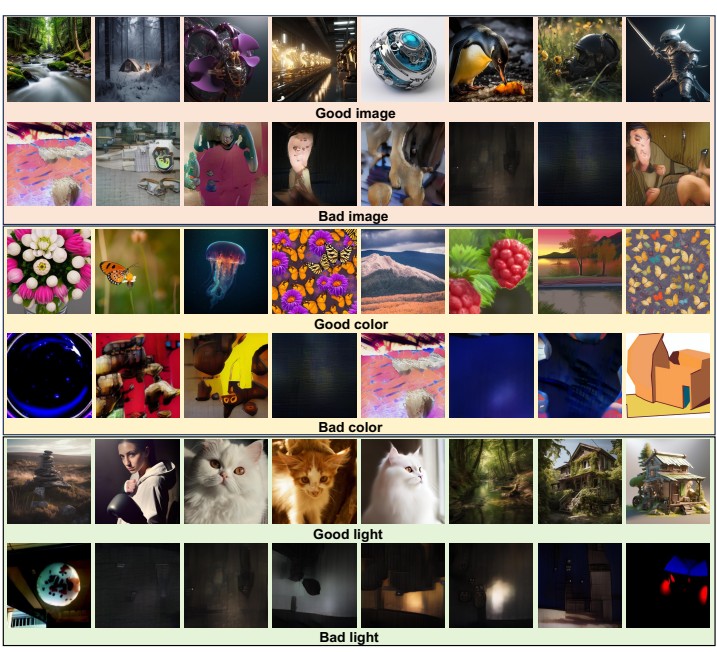

Figure 12: More image retrieval results with various text as queries on AGIQA-3K Li et al. (2023b).

**Example 1**

**Captioning with MOS-based text guidance**

**MOS of image**: 8.82 → **Text level**: perfect

**Instruction**: What you are given is a excellent quality image. Evaluate image quality based on factors such as *sharpness, color balance, and noise level*.

**Output**: The image of the silver Canon camera is of high quality. The noise is minimal, and the brightness is well-balanced, making it an excellent representation of the camera. The composition of the image is well-executed, with the camera being the main focus, and the background is clean and uncluttered. Overall, the image showcases the camera's features and design effectively.

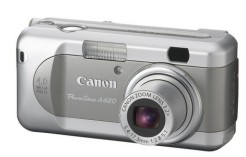

**Captioning without MOS-based text guidance**

**Instruction**: Evaluate image aesthetics based on factors such as *content, color, lighting, and composition*.

**Output**: The image quality of the silver digital camera is quite good. The camera appears to be sharp, with clear details visible in the image. The color balance is also well-maintained, with vibrant colors and accurate representation of the scene. Overall, the image quality of the camera is impressive, making it suitable for capturing high-quality photos and videos.

---

**Example 2**

**Captioning with MOS-based text guidance**

**MOS of image**: 3.91 → **Text level**: bad

**Instruction**: What you are given is a bad quality image. Evaluate image quality based on factors such as *sharpness, color balance, and noise level*.

**Output**: The camera appears to be in focus, but the overall quality of the image is poor. The brightness of the image is also low, which further contributes to the poor quality. The composition of the image is simple, with the camera being the main subject, and there are no other objects or elements in the frame.

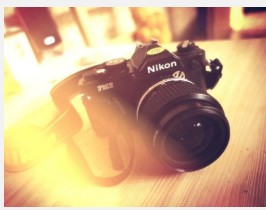

**Captioning without MOS-based text guidance**

**Instruction**: Evaluate image aesthetics based on factors such as *content, color, lighting, and composition*.

**Output**: The camera appears to be in good condition, with no visible signs of damage or wear. The image quality seems to be quite good, as the camera appears to be sharp and well-focused. The color balance appears to be accurate, with no noticeable color distortions or over-saturation. Overall, the image quality seems to be quite good, with a well-focused, accurate color balance, and minimal noise.

---

**Example 3**

**Captioning with MOS-based text guidance**

**MOS of image**: 3.65 → **Text level**: bad

**Instruction**: What you are given is a bad quality image. Evaluate image quality based on factors such as *sharpness, color balance, and noise level*.

**Output**: The image quality is poor, with a low resolution that makes it difficult to discern details. The brightness of the image is also low, making it challenging to see the subject clearly. The composition of the image is somewhat interesting, featuring a Chinese-style roof structure with a sign on it, but the overall quality of the image is not satisfactory.

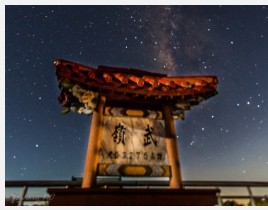

**Captioning without MOS-based text guidance**

**Instruction**: Evaluate image aesthetics based on factors such as *content, color, lighting, and composition*.

**Output**: The image quality appears to be quite good, with a sharp focus on the yellow and white sign. The colors in the image are well-balanced, and there is minimal noise, making it a visually appealing scene. The night sky in the background adds a sense of depth and contrast to the image, further enhancing its overall quality.

---

**Example 4**

**Captioning with MOS-based text guidance**

**MOS of image**: 1.01 → **Text level**: poor

**Instruction**: What you are given is a poor quality image. Evaluate image quality based on factors such as *sharpness, color balance, and noise level*.

**Output**: The image is of a concert with a group of people on stage. The resolution is low, which makes it difficult to discern the details of the performers and the audience. The brightness of the image is also low, making it hard to see the performers and the audience clearly. Overall, the image quality is poor due to low resolution, high noise, low brightness, and an unbalanced composition.

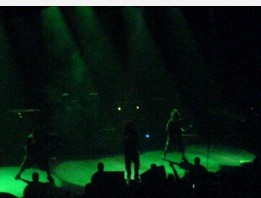

**Captioning without MOS-based text guidance**

**Instruction**: Evaluate image aesthetics based on factors such as *content, color, lighting, and composition*.

**Output**: The image quality appears to be quite good, with a high level of sharpness, color balance, and minimal noise. The image is a black and white photo of a band performing on stage, and the band members are clearly visible. The lighting in the scene is green, which adds a unique atmosphere to the image. Overall, the photo captures the essence of the live performance and the audience's engagement with the band.

Figure 13: Example of captioning with/without text guidance. The red text refers to MOS-based text guidance. The orange text highlights the correct quality-related caption. The green text indicates the incorrect quality-related caption without text guidance.

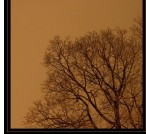 The Image

**(1) IR strategy**

1. I really like this photo. I love the see-through globe. I like that the colours are so close to the ones used in the map below. It's a great composition.
2. Cool idea. I think the constellations should have been the focal point, and blur the earth globe a little bit. But great idea!
3. use a polarizer next time to get rid of the reflection... or wait.. I hear they don't work on plastic.
4. I can undesrtand it as a zodiac shoot. Perhaps I know so much about it.
5. Cool. how did it look with the constellations being the focus?
6. Nice composition, DOF is about perfect, good color. Nice job!
7. Not sure this meets the challenge.
8. Thanks for all your coments!
9. Creative.

**(2) AR strategy**

1. I really like this photo. I love the see-through globe. I like that the colours are so close to the ones used in the map below. It's a great composition.
2. Cool idea. I think the constellations should have been the focal point, and blur the earth globe a little bit. But great idea!
3. Cool. how did it look with the constellations being the focus?
4. Nice composition, DOF is about perfect, good color. Nice job!
5. I can undesrtand it as a zodiac shoot. Perhaps I know so much about it.
6. use a polarizer next time to get rid of the reflection... or wait.. I hear they don't work on plastic.
7. Thanks for all your coments!
8. Creative.
9. Not sure this meets the challenge.

**(3) AIR strategy**

1. I really like this photo. I love the see-through globe. I like that the colours are so close to the ones used in the map below. It's a great composition.
2. Cool idea. I think the constellations should have been the focal point, and blur the earth globe a little bit. But great idea!
3. Cool. how did it look with the constellations being the focus?
4. I can undesrtand it as a zodiac shoot. Perhaps I know so much about it.
5. use a polarizer next time to get rid of the reflection... or wait.. I hear they don't work on plastic.
6. Nice composition, DOF is about perfect, good color. Nice job!
7. Thanks for all your coments!
8. Not sure this meets the challenge.
9. Creative.

---

The Image

**(1) IR strategy**

1. Ok, I have looked at this long and hard, even asked my wife and both of us could not figure any connection with the zodiac. So tell me what it is.. huh?
2. I like the sharp detail of the bare tree, but the background looks very unnatural and the frame seems too heavy for the picture.
3. I am still trying to relate to the challenge... Not quite sure... But the picture itself is beautiful! Nice tones.
4. I'm not sure how this relates to the challenge. I don't dislike this shot..in fact the colors are very appealing.
5. Nice picture, very artistic and postcard like, but I fail to see who does it meet the challenge.
6. Which zodiac sign is the tree? Nice picture but I don't see how it fits the challenge.
7. Love the dark sepia, but missing the Zodiac.
8. Love the composition, color, and exposure.
9. What symbol is this for?
10. zodiac?

**(2) AR strategy**

1. I am still trying to relate to the challenge... Not quite sure... But the picture itself is beautiful! Nice tones.
2. Love the dark sepia, but missing the Zodiac.
3. I like the sharp detail of the bare tree, but the background looks very unnatural and the frame seems too heavy for the picture.
4. Nice picture, very artistic and postcard like, but I fail to see who does it meet the challenge.
5. Which zodiac sign is the tree? Nice picture but I don't see how it fits the challenge.
6. Love the composition, color, and exposure.
7. I'm not sure how this relates to the challenge. I don't dislike this shot..in fact the colors are very appealing.
8. Ok, I have looked at this long and hard, even asked my wife and both of us could not figure any connection with the zodiac. So tell me what it is.. huh?
9. zodiac?
10. What symbol is this for?

**(3) AIR strategy**

1. I am still trying to relate to the challenge... Not quite sure... But the picture itself is beautiful! Nice tones.
2. I like the sharp detail of the bare tree, but the background looks very unnatural and the frame seems too heavy for the picture.
3. Love the dark sepia, but missing the Zodiac.
4. Nice picture, very artistic and postcard like, but I fail to see who does it meet the challenge.
5. Ok, I have looked at this long and hard, even asked my wife and both of us could not figure any connection with the zodiac. So tell me what it is.. huh?
6. I'm not sure how this relates to the challenge. I don't dislike this shot..in fact the colors are very appealing.
7. Which zodiac sign is the tree? Nice picture but I don't see how it fits the challenge.
8. Love the composition, color, and exposure.
9. zodiac?
10. What symbol is this for?

Figure 14: The comments sorted using different strategies. With our proposed strategy, we can obtain aesthetically relevant and informative comments. The order of comments is determined by the corresponding strategy. Gray sentences indicate being filtered.

**Quality-related captioning**

**MOS of image**: 7.21 → **Text level**: good
**Instruction**: Giving you a good quality image. Describe the quality of an image based on visual descriptors such as *resolution, noise, brightness*.
**Output**: The image is of a green and yellow parrot sitting on a wooden perch. The bird appears to be looking at the camera, and its vibrant colors make it stand out in the scene. The image is well-lit, with good contrast and resolution, allowing for a clear view of the parrot's features.

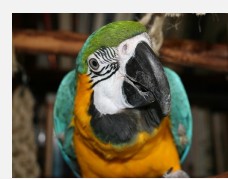

- - - - - - - - - - - - - - - - - - - - - - - - - - - - - - - - - - - - - - - - - - - - - - - - -

**MOS of image**: 7.80 → **Text level**: good
**Instruction**: What you are given is a good quality image. Evaluate image quality based on factors such as *sharpness, color balance, and noise levels*.
**Output**: The image of the elephant in the grassy field is of good quality. The resolution is high, allowing for clear details of the elephant's features, such as its tusks and trunk. The noise level is low, resulting in a clean and unobstructed view of the elephant. Overall, the image quality is excellent, making it an appealing representation of the elephant in its natural habitat.

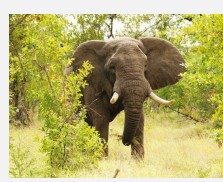

- - - - - - - - - - - - - - - - - - - - - - - - - - - - - - - - - - - - - - - - - - - - - - - - -

**MOS of image**: 2.73 → **Text level**: bad
**Instruction**: Giving you a bad quality image. Describe the quality of an image based on visual descriptors such as *resolution, noise, brightness*.
**Output**: The image quality is poor, with a low resolution that makes it difficult to discern details. The brightness is low, making it hard to see the pumpkin clearly. The noise in the image is high, which adds to the overall poor quality.

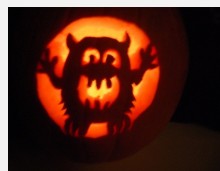

---

**Aesthetics-related captioning**

**MOS of image**: 5.09 → **Text level**: fair
**Instruction**: What you get is an image of fair aesthetics. Evaluate image aesthetics based on factors such as *content, color, lighting, and composition*.
**Output**: The image presents a visually appealing scene with a combination of content, color, lighting, and composition. The sunset in the background creates a warm and serene atmosphere. The combination of these elements results in an aesthetically pleasing image that captures the viewer's attention and evokes a sense of tranquility.

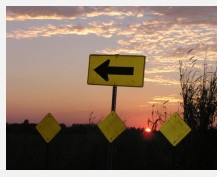

- - - - - - - - - - - - - - - - - - - - - - - - - - - - - - - - - - - - - - - - - - - - - - - - -

**MOS of image**: 3.05 → **Text level**: bad
**Instruction**: What you get is an image of bad aesthetics. Evaluate image aesthetics based on factors such as *content, color, lighting, and composition*.
**Output**: In the image, there is a house with a lit-up Christmas tree in front of it. However, the overall aesthetics of the image are not appealing due to a few factors. Firstly, the tree is positioned in the foreground, occupying a significant portion of the frame, which can make the image feel cramped and cluttered. Secondly, the tree's lights are not evenly distributed, with some areas appearing brighter than others. This uneven lighting can create an unbalanced and unappealing visual effect.

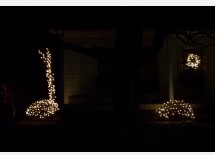

- - - - - - - - - - - - - - - - - - - - - - - - - - - - - - - - - - - - - - - - - - - - - - - - -

**MOS of image**: 8.04 → **Text level**: perfect
**Instruction**: What you get is an image of excellent aesthetics. Evaluate image aesthetics based on factors such as *content, color, lighting, and composition*.
**Output**: The image showcases an aesthetically pleasing scene of a lone tree standing in the middle of a grassy field at night. The tree is illuminated by the moonlight, creating a serene and peaceful atmosphere. The night sky is filled with stars, adding to the beauty and tranquility of the scene. The composition of the image, with the tree as the focal point, draws the viewer's attention and invites them to appreciate the natural beauty of the landscape. Overall, the image is visually appealing due to its content, color, lighting, and composition.

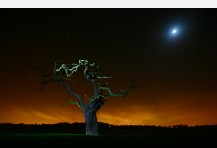

Figure 15: More examples of quality- and aesthetics-related captioning via MLLMs. The red text refers to MOS-based text guidance. The orange text highlights the quality- and aesthetics-related text.

