# OpenReview forum: "UniQA: Unified Vision-Language Pre-training for Image Quality and Aesthetic Assessment"
_ICLR.cc/2025/Conference — ICLR 2025 Conference Withdrawn Submission_

### Official Review · Reviewer_WcRe · 2024-10-31

**Soundness:** 3
**Presentation:** 3
**Contribution:** 3
**Rating:** 5
**Confidence:** 4

**Summary:**

This paper propose a Unified vision-language pre-training for Image Quality Assessment (IQA) and Image Aesthetic Assessment (IAA) tasks, which breaks down the barrier between quality-, and aesthetic-related features.  The authors construct a high-quality image-text dataset about image quality and aesthetics. Using their collected data, they develop UniQA, which learns a general perception of image assessment. Experiments show that the proposed method achieves SOTA performance across multiple IQA and IAA datasets.

**Strengths:**

1. The idea of developing a foundational model with robust visual assessment perceptions consistent with human to benefit both IQA and IAA tasks is novel.
2. The idea of using text descriptions as a bridge to integrate the two tasks is innovative
3. The paper provides a thorough evaluation of existing IQA and IAA models on respective IQA and IAA datasets.

**Weaknesses:**

1. The coarse image retrieval results in Fig. 4 may not sufficient. Adding the corresponding MOSs can increase its readability.
2. The assessment appear to be thorough and sound. However, the paper does not have any p-values or confidence intervals to support their comparisons of methods (especially for Tab. 1, 5, and 6).
3. The performance gain compared to other MOS regression-based models is limited, but introduce more efforts in collecting textual description, which poses challenge for the subjective study. I think this problem should be discussed further.

**Questions:**

1. Line41-42: “high-quality images tend to possess a higher aesthetic appeal compared to their low-quality counterparts.” This conclusion may not true since an aesthetically pleasing old photo may have some noise in it. Besides, some AI-generated images are over smooth which may exhibits high technical quality but with low aesthetic.
2. Is there any word limit in generating captions for images? Since the accuracy of description largely affect the performance.
3. What is the version of Qwen in Tab. 6 (Ablation on different MLLMs). Please include more information of parameter quantity. In addition, currently, the evaluated MLLMs are all open-source models, the performance difference between them is not significant, it is necessary to evaluate on proprietary MLLMs, such as GPT-4o, Gemini 1.5 Pro, and Qwen-VL-MAX, which owns better visual perception abilities. Experiments on Whether better description can enhance the performance is valuable.
4. When two images (for example, the right two good images of CLIVE in Fig. 4) have similar technical quality and semantic, the generated caption could be very close. In this case, the MOS-based text guidance may not reflect their true quality or aesthetics differences.
5. In line210-212: The author divide images into 5 levels based on MOS to obtain G, If an image’s MOS ranks in the top 20% of the score range, its level is assigned to perfect. The question is whether there is some loss in performance by quantifying the otherwise continuous scale scores back to a 5-level evaluation criterion, since the MOS distribution in the IQA dataset is almost inhomogeneous, sometimes showing left-skewed or right-skewed. In other words, does the granularity of the evaluation scale affect the model performance?
6. Another question is that the MOS of an image is collected from human study. The quality- and aesthetics-related captioning is completed by MLLMs. Whether the MLLMs can perceive as human remains an problem especially in some aesthetics judgments.
7. Using the length of a sentence as informativeness score may introduce large bias since it may contain some useless words. Why don’t you use Information relevance metrics to measure the informativeness of text?
8. It seems that the proposed method performs well on mainstream IQA datasets. The author further evaluate the generalization capability on AGIQA-3K. Since the naturalness problem [1] (consists of both technical quality and rationality (images which own similar technical quality with NSIs but could contain irrational contents) distortions) appears more easier in AI-generated images than pure quality problem. Is it possible to test on the AGIN dataset [1]? This may demonstrate a significant benefit of this method in the era of generative AI.

[1] Chen, Z., Sun, W., Wu, H., Zhang, Z., Jia, J., Ji, Z., ... & Zhang, W. (2023). Exploring the naturalness of ai-generated images. *arXiv preprint arXiv:2312.05476*.

---

> ### Author Response · Authors · 2024-11-25
> **Response to Reviewer WcRe [1/4]**
>
> We sincerely appreciate your helpful feedback. Your guidance is crucial in advancing our work. We have modified the paper based on your valuable comments, marked in red.
>
> > **Q1: The coarse image retrieval results in Fig. 4 may not sufficient. Adding the corresponding MOSs can increase its readability.**
>
> Thank you for your suggestion about the figure. We have added the MOS labels to the figure and have corrected the pdf. Please refer to Figure 4.
>
> ---
>
> > **Q2: The assessment appear to be thorough and sound. However, the paper does not have any p-values or confidence intervals to support their comparisons of methods (especially for Tab. 1, 5, and 6).**
>
> Thank you for your suggestion, we added the variance to reflect the confidence level of the results. For variance, we re-ran the experiments of CLIVE and KonIQ and calculate their variances. We compare the LIQE [1] method in the table below. Our method achieves close variance to LIQE. We will further supplement the full variance information in the future.
>
> | Method |CLIVE SRCC variances | CLIVE PLCC variances | KonIQ PLCC variances | KonIQ PLCC variances |
> | --- | --- | --- | --- | --- |
> | LIQE |  2e-4| 1.7e-4  | 1.6e-5|4e-6|
> | Ours | 3.2e-4 | 1.1e-4 |2.1e-5|7.3e-6|
>
>
> [1] Zhang W, Zhai G, Wei Y, et al. Blind image quality assessment via vision-language correspondence: A multitask learning perspective[C]//Proceedings of the IEEE/CVF conference on computer vision and pattern recognition. 2023: 14071-14081.
>
> ---
>
> > **Q3: The performance gain compared to other MOS regression-based models is limited, but introduce more efforts in collecting textual description.**
>
> We achieve highly competitive results on 7 IQA and 2 IAA datasets. Especially on the TID2013 dataset and CSIQ dataset, we achieve significant improvements compared with other methods, with SRCC values of 0.916 (v.s. 0.892 of DEIQT) and 0.963 (v.s. 0.947 of Re-IQA) respectively. Compared with other papers, such as LoDa (CVPR2024), it achieves 0.869 SRCC, even worse than previous methods (0.892 of DEIQT, AAAI2023). Furthermore, our method achieves impressive results on few-shot image evaluation, achieving the SRCC values of 0.828 (vs. 0.764 on CLIVE of GRepQ) and 0.844 (vs. 0.812 on KonIQ of GRepQ). In addition, our adapter fine-tuning (freeze UniQA backbone) is efficient and only requires 0.26M parameters to achieve these excellent results. In the future, we consider using more high-quality image-text data to improve the pre-training effect of UniQA.
>
> ---
>
> > **Q4: “high-quality images tend to possess a higher aesthetic appeal compared to their low-quality counterparts.” This conclusion may not true.**
>
> Thank you for your suggestion to improve our paper. We think your statement is quite correct. High-quality images do not necessarily have high aesthetic appeal. We also would like to clarify that high-quality sharp images are generally more advantageous than blurry images in terms of aesthetic appeal. Therefore, we have modified the PDF and marked it in red. The modified text is: "such that high-quality images **are more likely to** possess a higher aesthetic appeal compared to their low-quality counterparts". This makes the expression more reasonable and comprehensive.
>
> ---
>
> > **Q5: Is there any word limit in generating captions for images?**
>
> We use MOS-based text level guidance and task-specific content guidance to help MLLM generate reasonable text descriptions. Specifically, we first classify images into five text levels (bad, poor, fair, good, perfect) based on their MOS labels, so that there is a prompt: "This is a {level} quality image". For example, the images with a score of 78.2 will be classified as "good". This helps MLLM understand the quality of the image. Secondly, we prompt the MLLM with different words based on the data type, i.e., we use “Evaluate image quality based on factors such as sharpness, color balance, and noise level.” for IQA images and “Evaluate image aesthetics based on factors such as content, color, lighting, and composition.”. These two strategies help MLLM generate reasonable and meaningful text descriptions. We do not restrict the length of the text output by MLLM, as we found it difficult to get the MLLM to output sentences with a specific number of words.

---

> ### Author Response · Authors · 2024-11-25
> **Response to Reviewer WcRe [2/4]**
>
> > **Q6: What is the version of Qwen in Tab. 6 (Ablation on different MLLMs). It is recommended to use closed-source models such as GPT-4o to improve caption quality.**
>
> We use Qwen-v1-7B in Table 6. Please note that QWen-2-VL was not open source until after we completed the paper. Using GPT-4o to generate more than 300,000 image captions would cost nearly a thousand dollars. Considering economic issues, we do not choose gpt-4o to generate captions. In fact, from our experiments in Tab. 6 (Ablation on different MLLMs), using better MLLM for captioning does not bring significant performance improvements (e.g., on CLIVE, 0.871 SRCC using LLaVa-1.5-7B, 0.872 using LLaVa-1.5-7B, 0.870 using Qwen-v1-7B). This is because that MLLM usually generates texts with similar structures, which limits the diversity of the text dataset. In addition, we find that combining multiple MLLMs to generate diverse texts significantly enhance model performance. Therefore, we will consider integrating more MLLM or using in-context learning to improve text richness in the future.
>
> ---
>
> > **Q7: When two images have similar technical quality and semantic, the generated caption could be very close.**
>
> Even if two images have similar semantics and quality, MLLM can output text descriptions with different contents, which is very important for improving the diversity of the dataset. To demonstrate specifically, we here use LLaVa-7b to generate the caption of the right two good images of CLIVE in Figure 4. They are 1) “*The image is of high quality, with sharpness and color balance being the main factors contributing to its excellent quality. The clouds in the sky are well-defined, and the mountains in the background are crisp and clear. The overall color balance of the scene is well-maintained, with no noticeable color distortions or imbalances.*”; 2) “*The image appears to be of high quality. The sharpness of the image is evident, with clear details of the mountain, clouds, and surrounding landscape. The color balance is well-maintained, with vibrant colors and natural hues that accurately represent the scene.*”. We can observe that there are differences between the descriptions of the two images even though their semantics and quality are similar. **In addition**, we use MLLM to generate captions for more than 20,000 images. The sufficient amount of data reduces the possible harm of these similar images. **What's more**, we also use the captions of real human comments through data filtering. This also improves the diversity and effectiveness of our text data.
>
> ---
>
> > **Q8: Does the 5-level rating of images affect performance? Does the granularity of the evaluation scale affect the model performance?**
>
> We think that improve granularity of the evaluation scale (e.g., 10 text levels) will not lead to improved performance and may even confuse the MLLM, which is also not a typical practice. Firstly, we divide the images into 5 levels based on MOS labels. Using a specific text grade (a “good” image), to prompt the model is simpler and more direct than using a score (a ”78 / 100” score image, a high granularity method). This is due to the limited perceptual ability of MLLM for finer-grained image evaluation[1-2]. Secondly, many classic papers also choose to use 5 score levels to divide images [3-4]. What’s more, when humans annotate image quality/aesthetic scores, they also use a 5-level scale (i.e., 1 to 5 points) [5]. As a result, we empirically chose to use five levels to classify images.
>
> [1] Wu H, Zhang Z, Zhang E, et al. Q-Bench: A Benchmark for General-Purpose Foundation Models on Low-level Vision[C]//The Twelfth International Conference on Learning Representations.
>
> [2] Huang Y, Yuan Q, Sheng X, et al. Aesbench: An expert benchmark for multimodal large language models on image aesthetics perception[J]. arXiv preprint arXiv:2401.08276, 2024.
>
> [3] Series B. Methodology for the subjective assessment of the quality of television pictures[J]. Recommendation ITU-R BT, 2012, 500(13).
>
> [4] Zhang W, Zhai G, Wei Y, et al. Blind image quality assessment via vision-language correspondence: A multitask learning perspective[C]//Proceedings of the IEEE/CVF conference on computer vision and pattern recognition. 2023: 14071-14081.
>
> [5] Wu H, Zhang Z, Zhang W, et al. Q-align: Teaching lmms for visual scoring via discrete text-defined levels[J]. arXiv preprint arXiv:2312.17090, 2023.

---

> ### Author Response · Authors · 2024-11-25
> **Response to Reviewer WcRe [3/4]**
>
> > **Q9: Whether the MLLMs can perceive as human remains an problem especially in some aesthetics judgments.**
>
> MLLM has preliminary perception and judgment similar to that of humans. Firstly, MLLM is trained based on LLM and multimodal image-text data. The pre-training of LLM involves a large amount of unlabeled text written by humans. Multimodal image-text data are also labeled by humans. Therefore, the perception and understanding ability of MLLM is consistent with humans. Secondly, existing MLLM-based image evaluation papers also point out that MLLM has preliminary human quality and aesthetic evaluation perception [1][2]. In addition, to further guide MLLM to generate reliable image captions, we propose a MOS-guided task-specific prompt, which uses MOS prior information and evaluation factors (e.g., color balance, and noise level) to guide MLLM.
>
> [1] Wu H, Zhang Z, Zhang E, et al. Q-Bench: A Benchmark for General-Purpose Foundation Models on Low-level Vision[C]//The Twelfth International Conference on Learning Representations.
> [2] Huang Y, Yuan Q, Sheng X, et al. Aesbench: An expert benchmark for multimodal large language models on image aesthetics perception[J]. arXiv preprint arXiv:2401.08276, 2024.
>
>
> ---
>
> > **Q10: Why don’t you use Information relevance metrics?**
>
> Our approach has incorporated the Aesthetics-relevance Ranking, an effective metric for assessing both the quality of the text and its alignment with images. Consequently, we chose a straightforward method to reflect the text's informativeness by its sentence length. In fact, we have delved into a more effective technique in Appendix A.1, where we propose varying the weights assigned to Aesthetics-relevance (AR) and Image-relevance (IR) to better understand their influence on data quality in pre-training. For simplicity, we set the two weight factors to 1. The results from the ablation study in Table 6 (Ablation on data purification strategy), demonstrate that our strategy enhances model performance. In the future, if we collect more data, we will discuss how to assign the weights of the two strategies to further improve pre-training performance. More effective ways of measuring information are also worth exploring, such as Type Token Ratio (TTR), Distinct-n.

---

> ### Author Response · Authors · 2024-11-25
> **Response to Reviewer WcRe [4/4]**
>
> > **Q11: Is it possible to test on the AGIN dataset?**
>
> Thank you for your suggestion. We have evaluated our model on the AGIN dataset. The detailed results are shown in Appendix B.2 and Table 13. We report the results in the following table. We can observe that our method achieves competitive results on the AGIN dataset. Note that our method is not specifically designed for AI naturalness problems. These results demonstrate the strong generalization ability of our method. In addition, in Appendix B.2, we also test the effect of our method on large-scale AI generated IQA dataset AIGIQA-20K (Table 11) and an enhanced colonoscopy image quality assessment dataset ECIQAD (Table 12). Our model also achieves excellent results. This further demonstrates the generalization and effectiveness of our model.
>
> **Results on AGIN.** The * in table indicates that we also unfreeze the backbone for training with a smaller learning rate of 2e-6, which can achieve better performance. Our method can achieve the best top-2 performance, which is a competitive result.
>
> | Methods           | Technical SRCC | Technical PLCC | Rationality SRCC | Rationality PLCC | Naturalness SRCC | Naturalness PLCC |
> |--------------------|----------------|----------------|-------------------|-------------------|------------------|------------------|
> | BRISQUE           | 0.4867         | 0.4909         | 0.3608           | 0.3684           | 0.3745          | 0.4067          |
> | NIQE              | 0.4235         | 0.4279         | 0.3144           | 0.3211           | 0.3358          | 0.3378          |
> | DBCNN             | 0.7623         | 0.7661         | 0.6834           | 0.6838           | 0.7053          | 0.7128          |
> | HyperIQA          | 0.7752         | 0.7806         | 0.7196           | 0.7292           | 0.7365          | 0.7509          |
> | MUSIQ             | 0.7268         | 0.7355         | 0.6916           | 0.7013           | 0.7066          | 0.7139          |
> | UNIQUE            | 0.7358         | 0.7441         | 0.6934           | 0.6976           | 0.7104          | 0.7178          |
> | MANIQA            | 0.7763         | 0.7912         | 0.7192           | 0.7217           | 0.7355          | 0.7343          |
> | PAIAA             | 0.4763         | 0.4833         | 0.4532           | 0.4536           | 0.4453          | 0.4528          |
> | TANet             | 0.5882         | 0.6143         | 0.5037           | 0.4942           | 0.4948          | 0.4815          |
> | Del. Transf.      | 0.4299         | 0.4380         | 0.4009           | 0.4016           | 0.4196          | 0.4184          |
> | SAAN              | 0.8173         | 0.8235         | 0.7564           | 0.7711           | 0.7996          | 0.8028          |
> | JOINT             | 0.8351         | 0.8429         | 0.8033           | 0.8127           | 0.8264          | 0.8362          |
> | JOINT++           | **0.8351**     | **0.8429**     | **0.8033**       | **0.8127**       | **0.8264**      | **0.8362**      |
> | Ours              | 0.7524         | 0.8007         | 0.7728           | 0.7793           | 0.7882          | 0.7979          |
> | Ours*              | **0.7785**         | **0.8104**         | **0.7898**           | **0.7952**           | **0.8069**          | **0.8171**          |
>
> **Results on AIGIQA-20K.**
>
> | Method | SRCC | PLCC |
> | --- | --- | --- |
> | CLIPIQA | 0.331 | 0.483 |
> |CLIIQA+Finetune | 0.786 | 0.712|
> |CNNIQA  | 0.330 | 0.367|
> |CNNIQA  | 0.330 | 0.367|
> |Q-Align   | 0.746 | 0.742|
> |DBCNN    | 0.471 | 0.512|
> |DBCNN+Finetune    | 0.851| 0.869|
> |Ours  | 0.576 | 0.563 |
> |Ours+Finetune  | 0.830 | 0.885  |
> |Ours+Finetune* |**0.858** |**0.901** |
>
> **Results on ECIQAD.**
>
> | Method                           | SRCC   | PLCC   |
> |----------------------------------|--------|--------|
> | BRISQUE                      | 0.436  | 0.459  |
> | BIQME                       | 0.770  | 0.768  |
> | BPRI                        | 0.152  | 0.181  |
> | FRIQUEE                     | 0.663  | 0.656  |
> | CIQA                         | 0.738  | 0.735  |
> | ECIQ                        | 0.839  | 0.842  |
> | Ours                        | **0.873** | **0.887** |
> | Ours$^{*}$                   | **0.918** | **0.928** |

---

> ### Author Response · Authors · 2024-11-29
> **To Reviewer WcRe**
>
> Dear Reviewer WcRe,
>
> We truly appreciate your guidance to advance our work. We genuinely value the time and effort you dedicated to reviewing our paper. We are reaching out to ensure that our response adequately addressed all the questions and concerns you raised.
>
> Thank you for your valuable time, and we eagerly await your response.
>
> Best regards,
>
> Authors

---

> > ### Comment · Reviewer_WcRe · 2024-11-29
> >
> > Thank you for your detailed answers. Most of my concerns have been addressed. However, considering the actual innovation of such MLLM-based scoring strategy is minimal, I keep my rating at the borderline.

---

> ### Author Response · Authors · 2024-11-29
> **Response to Reviewer WcRe**
>
> Thank you for your timely response. Regarding your concerns about innovation, we have the following response:
>
> > **Q1 : considering the actual innovation of such MLLM-based scoring strategy is minimal.**
>
> - We want to clarify that **our work focuses on unified pre-training for joint IQA and IAA tasks to benefit various image assessment tasks.** To achieve this, we use MLLM to generate text with well-designed MOS-guided task-specific prompts and use the generated text to help us with data purification. Experiments show that the pre-trained model is beneficial for both IQA and IAA tasks. This provides inspiration for future researchers to create a more unified and universal image assessment model.
> - We propose prompt strategies and data purification strategies to help MLLM generate correct text and purify data. We propose a **MOS-guided task-specific prompt** to effectively guide MLLM generate correct description. Using MOS as a condition to control LMM to generate quality-related captions is innovative and meaningful. We introduce a simple yet effective Aesthetics-relevance and Informativeness Rank (AIR) to purify data. The work on dataset construction is a highlight of this paper.
> - **Our pre-trained model can be applied to various image assessment scenarios, including full supervision, zero-shot, few-label, image-text retrieval and other downstream image assessment tasks.** For example, UniQA can be effectively applied to AIGC image quality assessment, AIGC Image Naturalness assessment, and medical image assessment and other realistic scenarios. Therefore, our model has excellent generalization ability that can have beneficial effects on other image assessment tasks.

---

> > ### Author Response · Authors · 2024-12-03
> > **To Reviewer WcRe**
> >
> > Dear Reviewer WcRe,
> >
> > We truly appreciate your guidance to advance our work. We genuinely value the time and effort you dedicated to reviewing our paper. We are reaching out to ensure that our response adequately addressed all the questions and concerns you raised.
> >
> > We will open source this large-scale AI-generated text dataset on image quality and aesthetics. We believe that this will be useful for IQA and IAA methods based on multimodal learning. In addition, our method shows excellent performance in the field of AIGC image (Table 4, Table 11 and Table 13), which is also helpful for the future field of AIGC image quality assessment. In addition, our method can also be generalized to the field of medical image assessment (Table 12). In summary, our method can contribute to the field of image assessment.
> >
> > Thank you for your valuable time, and we eagerly await your response.
> >
> > Best regards,
> >
> > Authors

---

### Official Review · Reviewer_M7oA · 2024-11-02

**Soundness:** 2
**Presentation:** 2
**Contribution:** 2
**Rating:** 5
**Confidence:** 5

**Summary:**

To learn mutually beneficial representations shared by both tasks explicitly, this paper proposes Unified vision-language pre-training of Quality and Aesthetics (UniQA), which can extract useful and common representations from two tasks. Specifically, an image-text dataset about image quality and aesthetics is constructed by MLLMs, which is used to train the UniQA based on contrastive learning. Then a lightweight adapter is used to fine-tune the specific dataset of two tasks.

**Strengths:**

1.This paper includes extensive visualization experiments that provide a clear and intuitive demonstration of the experimental results of the proposed method.

2.Addressing the current scarcity of high-quality text-image datasets in IAA and the low quality of text descriptions, this study uses MLLM for data cleaning, constructing a higher-quality text-image IAA dataset to support future training.

**Weaknesses:**

1.Although the premise of this paper is intriguing (aiming to address both IQA and IAA problems simultaneously), similar approaches have already been proposed and with better outcomes. For example, [1] tackles IQA and IAA as well as VQA, surpassing this paper in scope and effectiveness. Additionally, this paper lacks a comparison with [1], making its experimental results less convincing.

2.The proposed method lacks substantial innovation. Overall, UniQA uses CLIP and a LoRA-like fine-tuning approach, with minimal improvements in training paradigms or network architecture.

3.Due to limited innovation in the training method, this work resembles more of a dataset paper, as its main contribution is the MLLM-based text-image IQA and IAA dataset. While the authors dedicate considerable detail to the dataset construction process, they fail to provide specifics on dataset content (e.g., image sources, dataset size, data distribution).

4.The writing structure is somewhat disorganized. For instance, in the introduction, the authors first introduce their UniQA method, then critique existing work, and return to explaining their method, which disrupts the flow. In the related work section, each subsection is extremely brief; more comprehensive discussion of recent work and analysis of similarities and distinctions between this method and others are needed. Overall, the writing structure should be refined to improve readability.

5.The experiments are insufficiently comprehensive. The comparison covers only a single 2024 method, which is far from adequate. As far as I know, several new methods were introduced in 2024, such as [1], [2], [3], [4], [5], etc. Additionally, it would be beneficial if Table 1 reported variance for the experimental results.

[1] Wu H, Zhang Z, Zhang W, et al. Q-align: Teaching lmms for visual scoring via discrete text-defined levels[J]. arXiv preprint arXiv:2312.17090, 2023.

[2] Zhong Y, Wu X, Zhang L, et al. Causal-IQA: Towards the Generalization of Image Quality Assessment Based on Causal Inference[C]//Forty-first International Conference on Machine Learning.2024

[3] Avanaki N J, Ghildiyal A, Barman N, et al. LAR-IQA: A Lightweight, Accurate, and Robust No-Reference Image Quality Assessment Model[J]. arXiv preprint arXiv:2408.17057, 2024.

[4] Yu Z, Guan F, Lu Y, et al. Sf-iqa: Quality and similarity integration for ai generated image quality assessment[C]//Proceedings of the IEEE/CVF Conference on Computer Vision and Pattern Recognition. 2024: 6692-6701.

[5] Fu H, Wang Y, Yang W, et al. DP-IQA: Utilizing Diffusion Prior for Blind Image Quality Assessment in the Wild[J]. arXiv preprint arXiv:2405.19996, 2024.

**Questions:**

1.The meaning of "versatile cues" in Line 99 is unclear. Where in the Methods section is this concept demonstrated? Additionally, what exactly does "Multi-Cue" refer to in Line 111?

2.What is the motivation for using sentence length as a score to represent information quantity in Lines 254–255?

---

> ### Author Response · Authors · 2024-11-25
> **Response to Reviewer M7oA [1/3]**
>
> We sincerely appreciate your helpful feedback. Your guidance is crucial in advancing our work. We have modified the paper based on your valuable comments, marked in red.
>
> > **Q1: Comparison and difference with Q-Align.**
>
> - **Performance comparison with Q-Align**. We have added the comparison results of Q-Align [1] to Table 1. Note that Q-Align only tests on the KonIQ and SPAQ, and does not repeat 10 times with random data split to take the median value. Therefore, we report the results from [2] (another paper of Q-Align team), which tests more datasets and has the same settings as ours.
>
> - **Difference with Q-Align**. Although both Q-Align and our method are pre-trained on multiple evaluation tasks, our method has differences and unique advantages. **Firstly**, Q-Align uses LLM, so the parameters is large (8.2B). In contrast, our method has only 0.15B parameters. Our fine-tuning is efficient, which can achieve competitive results by only training the adapter. **In addition**, UniQA can be regarded as an evaluation-aware CLIP, which can be used for various tasks, including full supervision IQA and IAA, zero-shot IQA, few-label IQA, and quality-related image-text retrieval. **More importantly**, we also supplement the generalization experiments of three datasets in Appendix B.2, including the enhanced colonoscopy image quality assessment dataset (ECIQAD), the AIGC Image Naturalness (AGIN) dataset, and the large-scale AIGC dataset AIGIQA20K. In these scenarios, our model also achieves excellent results, demonstrating the generalization ability and effectiveness of UniQA.
>
> [1] Wu H, Zhang Z, Zhang W, et al. Q-align: Teaching lmms for visual scoring via discrete text-defined levels[J]. arXiv preprint arXiv:2312.17090, 2023.
>
> [2] Zhu H, Wu H, Li Y, et al. Adaptive Image Quality Assessment via Teaching Large Multimodal Model to Compare[J]. arXiv preprint arXiv:2405.19298, 2024.
>
> ---
>
> > **Q2: Lacks substantial innovation in the training paradigms or network architecture.**
>
> - **Training paradigms**: We want to clarify that our work focuses on unified pre-training for joint IQA and IAA tasks to benefit various image assessment tasks. To achieve this, we use MLLM to generate text with well-designed MOS-guided task-specific prompts and use the generated text to help us with data purification. CLIP and contrastive learning are classic multimodal model and pre-training method, so we follow this pre-training pipeline.
> - **Fine-tuning approach and network architecture**. On the adapter, compared to other CLIP fine-tuned IQA works [1], we use more prompts (i.e., 5 text levels instead of just “good image”) to comprehensively evaluate image quality. We use a prompt ensemble strategy to fully utilize the knowledge of the pre-trained model to improve the model's ability in zero-shot and few-label evaluation.
>
> [1] Wang J, Chan K C K, Loy C C. Exploring clip for assessing the look and feel of images[C]//Proceedings of the AAAI Conference on Artificial Intelligence. 2023, 37(2): 2555-2563.
>
> ---
>
> > **Q3: Lack of detailed introduction to the dataset.**
>
> We supplement a detailed introduction to the dataset in Appendix C, marked in red, including the data volume of IQA and IAA, the length distribution of text sentences, and word clouds. In summary, we use FLIVE (IQA dataset, 39,807 images) and AVA (IAA dataset, 234,090 images) to generate a total of 1,240,915 text descriptions. The text length is concentrated in 20-30 words. The word cloud shows that it can be seen that the most common words in the text dataset are aesthetic and quality-related words, such as “aesthetics”, “quality”, “composition”, etc. This indicates that the text of the constructed dataset focuses on image assessment. Please refer to Appendix C and Figures 6, 7, and 8 for details.

---

> ### Author Response · Authors · 2024-11-25
> **Response to Reviewer M7oA [2/3]**
>
> > **Q4: The writing structure is somewhat disorganized. More comprehensive discussion of recent works are needed in the related work section.**
>
> - **Writing structure:** In the introduction, we first introduce the motivation (paragraph 1-2), i.e, unifying IQA and IAA tasks to develop a foundational model, then point out the shortcomings of existing methods in this goal and then propose UniQA (paragraph 3). Later, our paper focuses on why MOS cannot be used for pre-training directly (paragraph 4) and how to achieve multi-modal pre-training in the field of image assessment (paragraph 5).
> - **Related work:** We have added more comparison methods [1-7] to the related work section and compared and discussed the shortcomings of other unified IQA and IAA methods, such as Q-Align; in the IAA section, we also discussed the shortcomings of existing multimodal methods. We have revised the paper and marked it in red.
>
> [1] Xu K, Liao L, Xiao J, et al. Boosting Image Quality Assessment through Efficient Transformer Adaptation with Local Feature Enhancement[C]//Proceedings of the IEEE/CVF Conference on Computer Vision and Pattern Recognition. 2024: 2662-2672.
>
> [2] Shin N H, Lee S H, Kim C S. Blind Image Quality Assessment Based on Geometric Order Learning[C]//Proceedings of the IEEE/CVF Conference on Computer Vision and Pattern Recognition. 2024: 12799-12808.
>
> [3] Saha A, Mishra S, Bovik A C. Re-iqa: Unsupervised learning for image quality assessment in the wild[C]//Proceedings of the IEEE/CVF conference on computer vision and pattern recognition. 2023: 5846-5855.
>
> [4] Wu H, Zhang Z, Zhang W, et al. Q-align: Teaching lmms for visual scoring via discrete text-defined levels[J]. arXiv preprint arXiv:2312.17090, 2023.
>
> [5] Nie X, Hu B, Gao X, et al. BMI-Net: A Brain-inspired Multimodal Interaction Network for Image Aesthetic Assessment[C]//Proceedings of the 31st ACM International Conference on Multimedia. 2023: 5514-5522.
>
> [6] Huang Y, Li L, Chen P, et al. Coarse-to-fine Image Aesthetics Assessment With Dynamic Attribute Selection[J]. IEEE Transactions on Multimedia, 2024.
>
> [7] He S, Ming A, Zheng S, et al. Eat: An enhancer for aesthetics-oriented transformers[C]//Proceedings of the 31st ACM International Conference on Multimedia. 2023: 1023-1032.
>
> ---
>
> > **Q5: Add more comparison methods, add variance.**
>
> - **Add more comparison methods.** We have added these methods for comparison as you suggested, please refer to the revised PDF. Due to page limitations, we add these methods to Table 15 in the Appendix. Note that Sf-iqa [1] is different from the way we use AGIQA-3K. We test the zero-shot performance of UniQA on AGIQA-3K, and Sf-iqa is fully supervised, so we cannot compare. We add Sf-iqa to the related work section.
> - **Add variance**. For variance, we re-ran the experiments of CLIVE and KonIQ and calculated their variances. We compare the LIQE [2] method in the table below. Our method achieves close variance to LIQE. We will further supplement the full variance information in the future.
>
> | Method |CLIVE SRCC variances | CLIVE PLCC variances | KonIQ PLCC variances | KonIQ PLCC variances |
> | --- | --- | --- | --- | --- |
> | LIQE |  2e-4| 1.7e-4  | 1.6e-5|4e-6|
> | Ours | 3.2e-4 | 1.1e-4 |2.1e-5|9.3e-6|
>
> [1] Yu Z, Guan F, Lu Y, et al. Sf-iqa: Quality and similarity integration for ai generated image quality assessment[C]//Proceedings of the IEEE/CVF Conference on Computer Vision and Pattern Recognition. 2024: 6692-6701.
> [2] Zhang W, Zhai G, Wei Y, et al. Blind image quality assessment via vision-language correspondence: A multitask learning perspective[C]//Proceedings of the IEEE/CVF conference on computer vision and pattern recognition. 2023: 14071-14081.
>
> ---
>
> > **Q6: The meaning of "versatile cues" is unclear. Where in the Methods section is this concept demonstrated?**
>
> The “multi-cue” means that we use more prompts to evaluate an image, that is, {bad, poor, fair, good, perfect}. CLIPIQA [1] propose to use "good image" and "bad image" as anchors for multimodal image evaluation. Our proposed adapter uses 5 levels (i.e., multi-cue) of prompts to more comprehensively evaluate image quality. We have modified Section 4.2 to further clearly indicate the meaning of “multi-cue”, marked in red.
>
> [1] Wang J, Chan K C K, Loy C C. Exploring clip for assessing the look and feel of images[C]//Proceedings of the AAAI Conference on Artificial Intelligence. 2023, 37(2): 2555-2563.

---

> ### Author Response · Authors · 2024-11-25
> **Response to Reviewer M7oA [3/3]**
>
> > **Q7: What is the motivation for using sentence length as a score to represent information quantity?**
>
> Because when only aesthetic relevance is used, "good image" can also get a higher score. However, these rough image comments are harmful to the image-text alignment of CLIP, and thus we need more detailed descriptions. Therefore, aesthetic relevance is used to filter out comments that are not related to aesthetics. Text information helps those comments with detailed descriptions score higher. From the ablation experiment in Table 6, we notice that our two data purification strategies both improve the model, and the effect is best when they are used together, e.g., 0.876 to 0.890 SRCC on LIVEC with our AIR strategy.

---

> ### Author Response · Authors · 2024-11-29
> **To Reviewer M7oA**
>
> Dear Reviewer M7oA,
>
> We truly appreciate your guidance to advance our work. We genuinely value the time and effort you dedicated to reviewing our paper. We are reaching out to ensure that our response adequately addressed all the questions and concerns you raised.
>
> Thank you for your valuable time, and we eagerly await your response.
>
> Best regards,
>
> Authors

---

> ### Author Response · Authors · 2024-12-01
> **To Reviewer M7oA**
>
> Dear Reviewer M7oA,
>
> We truly appreciate your guidance to advance our work. We genuinely value the time and effort you dedicated to reviewing our paper. Considering that the discussion will end soon, we eagerly look forward to your response.
>
> Best regards,
>
> Authors

---

> > ### Comment · Reviewer_M7oA · 2024-12-02
> >
> > Thanks for the reviewer's response, however, upon consideration, I maintain my original score due to my continuing belief that the innovative aspect is insufficient.

---

> > > ### Author Response · Authors · 2024-12-03
> > > **Response to Reviewer M7oA**
> > >
> > > Thanks for your further response. Regarding your concerns about innovation, we have the following response:
> > >
> > > > **The innovation of this paper.**
> > >
> > > - We want to clarify that **our work focuses on unified pre-training for joint IQA and IAA tasks to benefit various image assessment tasks.** To achieve this, we use MLLM to generate text with well-designed MOS-guided task-specific prompts and use the generated text to help us with data purification. Experiments show that the pre-trained model is beneficial for both IQA and IAA tasks. This provides inspiration for future researchers to create a more unified and universal image assessment model.
> > > - We propose prompt strategies and data purification strategies to help MLLM generate correct text and purify data. We propose a **MOS-guided task-specific prompt** to effectively guide MLLM generate correct description. Using MOS as a condition to control LMM to generate quality-related captions is innovative and meaningful. We introduce a simple yet effective Aesthetics-relevance and Informativeness Rank (AIR) to purify data. The work on dataset construction is a highlight of this paper.
> > > - **Our pre-trained model can be applied to various image assessment scenarios, including full supervision, zero-shot, few-label, image-text retrieval and other downstream image assessment tasks.** For example, UniQA can be effectively applied to AIGC image quality assessment, AIGC Image Naturalness assessment, and medical image assessment and other realistic scenarios. Therefore, our model has excellent generalization ability that can have beneficial effects on other image assessment tasks.

---

> > > > ### Author Response · Authors · 2024-12-03
> > > > **To Reviewer M7oA**
> > > >
> > > > Dear Reviewer M7oA,
> > > >
> > > > We truly appreciate your guidance to advance our work. We genuinely value the time and effort you dedicated to reviewing our paper. We are reaching out to ensure that our response adequately addressed all the questions and concerns you raised.
> > > >
> > > > We will open source this large-scale AI-generated text dataset on image quality and aesthetics. We believe that this will be useful for IQA and IAA methods based on multimodal learning. In addition, our method shows excellent performance in the field of AIGC image (Table 4, Table 11 and Table 13), which is also helpful for the future field of AIGC image quality assessment. In addition, our method can also be generalized to the field of medical image assessment (Table 12). In summary, our method can contribute to the field of image assessment.
> > > >
> > > > Thank you for your valuable time, and we eagerly await your response.
> > > >
> > > > Best regards,
> > > >
> > > > Authors

---

### Official Review · Reviewer_vpBA · 2024-11-03

**Soundness:** 3
**Presentation:** 3
**Contribution:** 2
**Rating:** 6
**Confidence:** 4

**Summary:**

This paper introduces a unified vision-language pre-training of quality and aesthetics (UniQA) to tackle both the image quality assessment (IQA) and the image aesthetic assessment (IAA) tasks.
The proposed method first generates quality- and aesthetics-related descriptions by using multimodal large language model (MLLMs) and use them to refine authentic noisy data.
Then the UniQA model is pre-trained with the purified data and finally a lightweight adapter is adapted to each IQA and IAA benchmark.
In the pre-training, two tasks are bridged and the UniQA can learn rich correlated information to enhance the both tasks.

**Strengths:**

- Effective data generation and refinement using MLLMs and efficient adaptation to each benchmark.
- Tackles IQA and IAA at the same time.
- Comparisons with a number of previous methods.

**Weaknesses:**

- In obtaining IR score, the informativeness of text is measured by the sentence length but it seems sub-optimal and less robust.
- It is not verified that the quality-level keywords, "bad, poor, fair, good, perfect", are reasonable without any intuitions or experiments. In addition, they are linked to score values (fig 3 b) "0.2, 0.4, 0.6, 0.8, 1.0", respectively, but it is an arbitrary matching. Likewise, the reason using a prompt ensemble with keywords "extremely blurry, blurry, fair, sharp, extremely sharp" have not been verified.
- The results of the prompt ensemble, which gives a major improvement, are not reported in table 1-3.
- It is unclear how can the UniQA handle real-world images.

**Questions:**

- How the MOS-based text guidance G is obtained? Have the authors try several other attempts?
- What exactly does multi-cue mean in the multi-cue integration adapter.
- How much will performance improve if replace the backbone CLIP-B/16 to other MLLMs such as LLaVA or more latest models? If it improves then why not use it?

---

> ### Author Response · Authors · 2024-11-25
> **Response to Reviewer vpBA [1/2]**
>
> We sincerely appreciate your helpful feedback. Your guidance is crucial in advancing our work. If you have any further questions, please feel free to let us know.
>
> > **Q1: The IR score is sub-optimal and less robust.**
>
> We have used Aesthetics-relevance Rank, which can well describe the quality of text and the degree of text-image matching. Therefore, we simply used sentence length to measure the information of the text. In fact, we have discussed a more effective and reasonable method in Appendix A.1. We can assign different weights to AR and IR to further explore the impact of data quality on pre-training. Considering the simplicity, we set the two factors to 1. From the ablation experiment Table 6, we can see that our strategy can bring performance improvement to the model. In the future, if we collect more data, we will discuss how to assign the weights of the two strategies to further improve pre-training performance. More effective ways of measuring information are also worth exploring, such as Type Token Ratio (TTR), Distinct-n.
>
> ---
>
> > **Q2: (1) The rationality of quality-level keywords. (2) Words and scores are matched arbitrarily. (3) The reason of using a prompt ensemble.**
>
> - Many classic papers choose to use these 5 text levels to divide images [1][2]. In addition, when humans annotate image quality/aesthetic scores, they also use a 5-level scale (i.e., 1 to 5 points) [3]. As a result, we empirically chose to use five levels to classify images.
> - These five words have a one-to-one correspondence with {0.2, 0.4, 0.6, 0.8, 1.0}, e.g., 0.2 is assigned to “bad” and 0.6 is assigned to “fair”. The prompt ensemble strategy typically uses multiple evaluation-related words to assess image respectively and take their average as the quality score.
> - Because a single set of evaluation words cannot summarize the image situation well, for example, “good image” may be too general, adding “noise-free image” and “sharp image” can better help the model understand what “good” means. Please see Table 10 for prompt ensemble details.
>
> [1] Series B. Methodology for the subjective assessment of the quality of television pictures[J]. Recommendation ITU-R BT, 2012, 500(13).
>
> [2] Zhang W, Zhai G, Wei Y, et al. Blind image quality assessment via vision-language correspondence: A multitask learning perspective[C]//Proceedings of the IEEE/CVF conference on computer vision and pattern recognition. 2023: 14071-14081.
>
> [3] Wu H, Zhang Z, Zhang W, et al. Q-align: Teaching lmms for visual scoring via discrete text-defined levels[J]. arXiv preprint arXiv:2312.17090, 2023.
>
> ---
>
> > **Q3: The effectiveness of prompt ensemble.**
>
> The prompt ensemble is effective for zero-shot and few-label image evaluation because it can evaluate the image more comprehensively. We report the results using prompt ensemble in Table 4 (zero-shot) and 5 (few-label). For example, using only 50 images (i.e., few-label setting) for training, prompt ensemble can significantly improve the results from 0.772 SRCC to 0.844 SRCC (Table 5). In zero-shot scenarios, it can also bring significant performance improvements (from 0.638 to 0.790 on LIVEC in Table 4). For the fully supervised fine-tuning (Table 1-3), since we use a trainable adapter, which can adjust the weights to adapt to different datasets, so the improvement is not obvious.
>
> ---
>
> > **Q4: It is unclear how can the UniQA handle real-world images.**
>
> UniQA can directly evaluate the real-world images in a zero-shot manner (Table 4). For example, we can directly use the cosine similarity score between the template "good image" and the image as the quality score. More comprehensively, we calculate the similarity between {bad, poor, fair, good, perfect} and the image and multiply them by {0.2, 0.4, 0.6, 0.8, 1.0} respectively, and finally sum them up as the quality score. In addition, the images of datasets in Table 1-3 are collected from real-world. The trained model on these datasets can also predict the quality of real-world images.
>
> ---
>
> > **Q5: (1) How the MOS-based text guidance G is obtained? (2) Have the authors try several other attempts?**
>
> - We divide images into 5 text levels based on MOS, i.e., {bad, poor, fair, good, perfect}. Specifically, image with MOS between 0 and 20 is assigned as “bad”, between 20 and 40 is assigned as “poor”, and so on.
>
> - Yes, we tried using MOS directly as prompts, for example, “a 78 / 100 score image” instead of “a good image”. However, we found that MLLM have poor understanding of this form of prompts and therefore cannot output the correct quality text description. We also tried to prompt MLLM without MOS-based text guidance and found that MLLM failed to generate correct descriptions, refer to Figure 13 for details.

---

> ### Author Response · Authors · 2024-11-25
> **Response to Reviewer vpBA [2/2]**
>
> > **Q6: What exactly does multi-cue mean in the multi-cue integration adapter.**
>
> The “multi-cue” means that we use more prompts to evaluate an image, that is, {bad, poor, fair, good, perfect}.  CLIPIQA [1] propose to use "good image" and "bad image" as anchors for multimodal image evaluation. Our proposed adapter uses 5 levels (i.e., multi-cue) of prompts to more comprehensively evaluate image quality. We have modified Section 4.2 to clearly indicate the meaning of “multi-cue”, marked in red.
>
> [1] Wang J, Chan K C K, Loy C C. Exploring clip for assessing the look and feel of images[C]//Proceedings of the AAAI Conference on Artificial Intelligence. 2023, 37(2): 2555-2563.
>
> ---
>
> > **Q7: How much will performance improve if replace the backbone CLIP-B/16 to other MLLMs such as LLaVA or more latest models? If it improves then why not use it?**
>
> The LLaVa model cannot directly replace CLIP. CLIP is a multimodal image-text alignment model that includes visual and text encoders to generate aligned features. LLaVa combines LLM and visual models for complex image-text interaction understanding and visual question answer. In addition, some works [1][2] have also pointed out that directly using a large multimodal model such as LLaVa for image evaluation cannot achieve satisfactory results. Moreover, previous multimodal image assessment methods [3][4] are also based on CLIP. For fairness, we don’t use improved versions of CLIP, such as Evaclip [5] and BLIP [6].
>
> [1] Wu H, Zhang Z, Zhang E, et al. Q-Bench: A Benchmark for General-Purpose Foundation Models on Low-level Vision[C]//The Twelfth International Conference on Learning Representations.
>
> [2] Huang Y, Yuan Q, Sheng X, et al. Aesbench: An expert benchmark for multimodal large language models on image aesthetics perception[J]. arXiv preprint arXiv:2401.08276, 2024.
>
> [3] Zhang W, Zhai G, Wei Y, et al. Blind image quality assessment via vision-language correspondence: A multitask learning perspective[C]//Proceedings of the IEEE/CVF conference on computer vision and pattern recognition. 2023: 14071-14081.
>
> [4] Wang J, Chan K C K, Loy C C. Exploring clip for assessing the look and feel of images[C]//Proceedings of the AAAI Conference on Artificial Intelligence. 2023, 37(2): 2555-2563.
>
> [5] Sun Q, Fang Y, Wu L, et al. Eva-clip: Improved training techniques for clip at scale[J]. arXiv preprint arXiv:2303.15389, 2023.
>
> [6] Li J, Li D, Xiong C, et al. Blip: Bootstrapping language-image pre-training for unified vision-language understanding and generation[C]//International conference on machine learning. PMLR, 2022: 12888-12900.

---

> ### Author Response · Authors · 2024-11-29
> **To Reviewer vpBA**
>
> Dear Reviewer vpBA,
>
> We truly appreciate your guidance to advance our work. We genuinely value the time and effort you dedicated to reviewing our paper. We are reaching out to ensure that our response adequately addressed all the questions and concerns you raised.
>
> Thank you for your valuable time, and we eagerly await your response.
>
> Best regards,
>
> Authors

---

> > ### Comment · Reviewer_vpBA · 2024-11-29
> >
> > Thanks for your detailed response, however, some concerns are not addressed well, especially Q2 and Q3.
> >
> > Q2) Human likely think (bad, poor, fair, good, perfect) to match (0.2, 0.4, 0.6, 0.8, 1.0) respectively is reasonable, however, the trained model may not. I have a concern that this linking seems just human's intuition not found by a kind of searching or considering model's weight. Likewise in Q3.
> >
> > Q4) So which adapter could we use? This is not a major concern, however, it makes the proposed method more practical and powerful if possible.
> >
> > Q7) It also not a major concern, but, it would better to show the proposed method can achieve a big step forward using Evaclip or BLIP even it is unfair.
> >
> > My raiting is still in between 5 and 6.

---

> ### Author Response · Authors · 2024-11-30
> **Response to Reviewer vpBA**
>
> We sincerely appreciate your further constructive feedback and comments. Regarding your concerns, our response is as follows:
>
> > **Q1: Human likely think (bad, poor, fair, good, perfect) to match (0.2, 0.4, 0.6, 0.8, 1.0) respectively is reasonable, however, the trained model may not.  I have a concern that this linking seems just human's intuition not found by a kind of searching or considering model's weight. Likewise in Q3.**
>
> First, the score level (0.2, 0.4, 0.6, 0.8, 1.0) here is a learnable parameter, which we do not point out in the paper (we will correct the paper). The model can adjust this parameter based on the training data and its own weight. **Therefore, this score level actually takes into account both human perception and model preference.** In addition, adapter is also learnable, and it can also adjust the weights based on the training data.
>
> > **Q2: So which adapter could we use? This is not a major concern, however, it makes the proposed method more practical and powerful if possible.**
>
> The adapter is mainly used to fine-tune when applying UniQA to a specific dataset. When using UniQA for zero-shot real-world image evaluation, there is no need to use an adapter. In this case, UniQA can be regarded as a image evaluation-aware CLIP. The image quality score is calculated by calculating the cosine similarity between the image and the text. For more accurate image evaluation, we can use a model that has fine-tuned UniQA and the adapter on a real dataset (such as KonIQ). This model (UniQA plus the adapter) can be directly used for real-world evaluation. The adapter here uses {bad, poor, fair, good, perfect} as the prompt and learnable {0.2, 0.4, 0.6, 0.8, 1.0} parameters as the score level.
>
> > **Q3: it would better to show the proposed method can achieve a big step forward using Evaclip or BLIP even it is unfair.**
>
> We use EVA-CLIP-B-16 to further perform experiments. Note that we found that BLIP model architecture is different from CLIP and cannot be directly used in our scenario. The experimental results are shown in the table below. We can observe that using EVA-CLIP does not bring obvious performance improvement. This is reasonable because EVA-CLIP only outperforms CLIP on natural scenes. However, similar progress cannot be made in the field of image evaluation. Therefore, using high-quality image quality and aesthetics-related image-text datasets for pre-training is the key to improving model performance. In the future, we will collect more data to further improve the effectiveness of model pre-training.
>
> **Comparison results with EVA-CLIP.**
> | Method                           | LIVEC SRCC   | KonIQ PLCC   | AVA PLCC   |
> |----------------------------------|--------|--------|--------|
> | CLIP (ours)                      | 0.890  | 0.933  |0.776  |
> | EVA-CLIP                       | 0.892  | 0.932  | 0.778  |

---

> > ### Comment · Reviewer_vpBA · 2024-12-02
> >
> > Q1) In my opinion, this is one of the strong features of the proposed method, however, this part is not properly explained in the paper and some readers may misunderstand. In addition, a relevant analysis should be provided, for example, score level values before and after the training and how those changes affect performance. I will raise my rating to 6 considering the learnable score level, but again still some parts could be improved to eliminate any potential for misunderstanding.

---

> ### Author Response · Authors · 2024-12-03
> **Response to Reviewer vpBA**
>
> Thanks for raising the score. We truly appreciate your recognition of our work, which encourages our further work.
>
> We will modify the paper to indicate that this score level is a learnable parameter to eliminate misunderstandings. We discuss the comparison of the model's score level before and after training. The original score level is {0.2, 0.4, 0.6, 0.8, 1.0}. After training, on the CLIVE dataset, the score level is {0.1353, 0.3997, 0.6026, 0.8032, 0.9950}; on the AGIQA3k dataset, the score level is {0.2655, 0.4265, 0.6072, 0.8427, 1.044}. We can observe that the score level is adaptively adjusted according to the dataset. On the CLIVE dataset, the score of "bad" changes from 0.2 to 0.135, indicating that the overall image quality of the CLIVE dataset is relatively good. On the AGIQA3k dataset, the score of "bad" changes from 0.2 to 0.2655. This indicates that there are many AI-generated low-quality images in AGIQA3k dataset. In addition, we find that datasets in different scenarios, such as CLIVE and AGIQA3k, have different patterns in score level. We will further discuss and analyze this issue and add these to the appendix.
>
> Thanks again for your constructive suggestions, which enhance the quality of our work.
>
> Best wishes

---

### Official Review · Reviewer_uVpG · 2024-11-03

**Soundness:** 3
**Presentation:** 3
**Contribution:** 2
**Rating:** 6
**Confidence:** 3

**Summary:**

This paper introduces a unified model for assessing both image quality and aesthetics, called UniQA. It uses multimodal large language models (MLLMs) to generate descriptive text for IQA and IAA datasets, which allows for a more detailed and refined training dataset. The model uses these captions for pre-training and employs a lightweight Multi-Cue Integration Adapter to adapt the model for downstream tasks. Experimental results show UniQA achieving state-of-the-art (SOTA) performance across multiple datasets.

**Strengths:**

1. UniQA effectively combines IQA and IAA tasks, extracting shared representations, leading to efficient and comprehensive visual assessment capabilities.

2. Using MLLMs to generate high-quality text descriptions enriches the dataset.

3. The lightweight Multi-Cue Integration Adapter allows UniQA to adapt efficiently to various downstream IQA and IAA tasks with minimal parameter adjustments.

**Weaknesses:**

1. Although this paper effectively utilizes MLLM-generated text for dataset construction, the generated descriptions tend to have similar structures and expressions, resulting in limited text diversity. The model's performance heavily depends on the quality of MLLM-generated text, which may introduce noise or bias, especially as MLLMs may produce overly positive or vague evaluations when generating image descriptions.

2. While this method aligns IQA and IAA datasets by unifying them on a common MOS scale to reduce MOS biases between datasets, such direct alignment may overlook the inherent rating standards and subtle differences across datasets. For instance, different datasets may prioritize specific visual features (e.g., sharpness, color balance) in quality assessment, while aesthetic assessment might focus more on composition and emotional impact. In such cases, a unified MOS scale may reduce the model's sensitivity to certain features, potentially compromising its performance precision in specific tasks.

3. The effectiveness of the dataset constructed in this paper largely depends on MLLM-generated text descriptions and data purification strategies. However, it lacks a systematic evaluation of dataset quality. While the data purification process uses Aesthetics-relevance Rank (AR) and Informativeness Rank (IR) to filter out irrelevant content, these subjective criteria may filter out some valuable information, which could affect the dataset's diversity and comprehensiveness.

**Questions:**

See Weakness

---

> ### Author Response · Authors · 2024-11-25
> **Response to Reviewer uVpG**
>
> We appreciate your valuable comments and recognition of our work. Your advice significantly helps in enhancing the quality of our work. If you have any further questions, please feel free to let us know.
>
> > **Q1: Effectiveness of MLLM-generated text, including text diversity and text quality.**
>
> - **Text diversity**. To improve text diversity, we try to generate text by integrating multiple MLLMs. As shown in Table 6 (Ablation on different MLLMs), this method can improve the model performance. In addition, we use real human-annotated aesthetic assessment comments, which can also improve the diversity and richness of the text.
> - **Text quality**. We propose the MOS-guided task-specific prompts to enable MLLM to generate correct fine-grained text descriptions. We show some examples in Appendix D and Figure 13. Without our proposed prompt strategy, MLLM cannot output correct and reasonable descriptions. The ablation experiment in Table 6 (Ablation on different pre-training data) also shows that pre-training with Y_IQA and Y_IAA generated by MLLM can improve the performance of the model. For example, using MLLM generated Y_IQA for training improves the KonIQ from 0.907 to 0.914 SRCC and AVA from 0.748 to 0.755 SRCC. This demonstrates the effectiveness of the generated text.
> ---
>
> > **Q2: MOS normalization may overlooks the inherent rating standards and subtle differences across datasets, e.g., specific visual features.**
> - **Overlooks the inherent rating standards**.  Classifying the image into 5 levels according to the MOS value can effectively guide the MLLM to generate correct text. Using a finer-grained prompt (a "78 / 100" score image) will not improve the text quality and may even confuse the MLLM. This is due to the poor perceptual ability of MLLM for finer-grained image evaluation[1-2].
>
> - **Overlooks the subtle differences across datasets, e.g., specific visual features.** For the visual features of different datasets, we propose the task-specific prompt to deal with them. For instance, we prompt the MLLM with “*evaluate image based on sharpness, color balance, and noise level*” for IQA datasets; and use “*content, color, lighting, and composition*” for IAA datasets. This approach helps MLLM focus on multiple different aspects of the dataset and thus generate comprehensive captions. We also use three different prompt for each image to further improve text diversity, as shown in Figure 9 in appendix.
>
> [1] Wu H, Zhang Z, Zhang E, et al. Q-Bench: A Benchmark for General-Purpose Foundation Models on Low-level Vision[C]//The Twelfth International Conference on Learning Representations.
>
> [2] Huang Y, Yuan Q, Sheng X, et al. Aesbench: An expert benchmark for multimodal large language models on image aesthetics perception[J]. arXiv preprint arXiv:2401.08276, 2024.
>
> ---
>
> > **Q3: Lacks a systematic evaluation of dataset quality; Data purification strategy may filter out some valuable information.**
>
> - **Evaluation of dataset quality**. Thank you for your constructive suggestions. We provide a detailed introduction to the dataset in Appendix C, including the data volume of IQA and IAA, the length distribution of text sentences, and word clouds. In summary, we use FLIVE (IQA dataset, 39,807 images) and AVA (IAA dataset, 234,090 images) to generate a total of 1,240,915 text descriptions. The text length is concentrated in 20-30 words. The word cloud shows that it can be seen that the most common words in the text dataset are aesthetic and quality-related words, such as “aesthetics”, “quality”, “composition”, etc. This indicates that the text of the constructed dataset focuses on image assessment. Please refer to Appendix C and Figure 6, 7, and 8 for details.
>
> - **Data purification strategy**. In fact, we have delved into a superior and more rational technique in Appendix A.1, where we propose varying the weights assigned to Aesthetics-relevance (AR) and Image-relevance (IR) to better understand their influence on data quality in pre-training. For simplicity, we set the two weight factors to  1. The ablation experiment in Table 6 (Ablation on data purification strategy) shows that this strategy has a positive impact on the model performance, e.g., 0.876 to 0.890 SRCC on CLIVE with our strategy.  In the future, if we collect more data, we will discuss how to assign the weights of the two strategies to further improve pre-training performance. More effective ways of measuring information are also worth exploring.

---

> > ### Comment · Reviewer_uVpG · 2024-12-02
> >
> > I thank the authors for their response. They have addressed all my questions and concerns. I will keep the score at 6.

---

### Official Review · Reviewer_6yW7 · 2024-11-03

**Soundness:** 3
**Presentation:** 3
**Contribution:** 3
**Rating:** 6
**Confidence:** 3

**Summary:**

This paper designs a novel UniQA framework to perform IQA tasks on different types of visual content. Contains two key designs: (1) Utilize MLLM to generate high-quality text descriptions; (2) Use the text generated for IAA as metadata to purify noisy IAA data. Experiments show that this method achieves state-of-the-art performance on both IQA and IAA tasks, achieving SRCC/PLCC above 0.9.

**Strengths:**

The refine module proposed in this paper can transform messy data into a quality-related format, thereby ensuring that the model is highly consistent with the supervised experience of the human eye, and has the potential to be applied in future IQA and IAA tasks. This can achieve the migration of large-scale, general descriptive datasets to quality-related specialized datasets, which can promote progress in the field of IQA.

The method proposed in this paper can be used for both traditional visual content and emerging AIGC quality assessment. It can promote the further application of AIGC, especially the aesthetic aspects of AIGC.

The experimental dataset is relatively sufficient, and the comparison method is complete and advanced.

**Weaknesses:**

The core implementation of UniQA is to predict the probability of The image quality is {bad, poor, fair, good, perfect} and then fuse them. This is not a completely new paradigm. As far as I know, this first appeared in Q-Align. However, the author only reviewed this article without comparing them in experiments. Considering the similarities between the two, it is necessary for the author to conduct comparative experiments in Table 1 and emphasize the differences between the two.

It is a good point that the UniQA proposed by the author has a significant advantage in evaluating AIGC. However, the verification using AGIQA-1K/3K is slightly outdated. T2I generation models are developing very rapidly, and even the best 3K images (June 2023) are only of average quality compared to the current AIGC. Therefore, the author can verify the performance of UniQA on the AIGIQA-20K (April 2024) database to better prove its applicability to AIGC.

**Questions:**

I am curious whether the combination of IQA and IAA pipelines makes sense. In the ablation experiment, using only IQA does not seem to be too bad. Therefore, I am not sure whether it is necessary to use a dedicated IAA pipeline. I think the quality in IQA is not purely low-level quality, but also includes aesthetic elements, so removing IAA will not lead to a significant drop in performance. Note that this is not to question weakness, just kindly discuss with the author: Is IAA a subtask of IQA, or an equally important task?

---

> ### Author Response · Authors · 2024-11-25
> **Response to Reviewer 6yW7 [1/3]**
>
> Thank you for your valuable comments and recognition! We have addressed each of the issues you raised and made the necessary revisions to our manuscript. The changes are marked in red.
>
> > **Q1: Compare with Q-Align in the experiment and emphasize the differences between the two.**
>
> - **Performance comparison with Q-Align.** We have added the comparison results of Q-Align [1] to Table 1. Note that Q-Align only tests on the KonIQ and SPAQ, and does not repeat 10 times with random data split to take the median value. Therefore, we report the results from [2] (another paper of Q-Align team), which tests more datasets and has the same settings as ours.
>
> - **Difference with Q-Align.** Both papers use five evaluation-related words to obtain quality scores. However, their specific usage is different. **Firstly**, Q-Align uses the logits of these five words of LLM to obtain the quality score. We use the cosine similarity between the five words and the image to weight the score level. **Secondly**, Q-Align uses LLM, so the parameters is large (8.2B). In contrast, our method has only 0.15B parameters. Our fine-tuning is efficient, which can achieve competitive results by only training the adapter. **In addition**, our method focuses on pre-training. UniQA can be used as an image assessment-aware CLIP for various evaluation-related downstream tasks (e.g., few-label IQA in Table 5, AIGC IQA in Table 11, medical Image IQA in Table 12, AI Naturalness evaluation in Table 13), which is a highlight of our method.
>
> [1] Wu H, Zhang Z, Zhang W, et al. Q-align: Teaching lmms for visual scoring via discrete text-defined levels[J]. arXiv preprint arXiv:2312.17090, 2023.
>
> [2] Zhu H, Wu H, Li Y, et al. Adaptive Image Quality Assessment via Teaching Large Multimodal Model to Compare[J]. arXiv preprint arXiv:2405.19298, 2024.

---

> ### Author Response · Authors · 2024-11-25
> **Response to Reviewer 6yW7 [2/3]**
>
> > **Q2: Further validating the model on the large-scale AIGC dataset AIGIQA20K.**
>
> We have verified our method on a larger AI generated image evaluation dataset AIGIQA20K. We supplement the results in Appendix B.2 and Table 11. We evaluate both fine-tuning and zero-shot settings. As shown in the table below, our method achieves competitive results in both settings, demonstrating the excellent generalization performance of our method on AI-generated images. What’s more, we use the enhanced colonoscopy image quality assessment dataset (ECIQAD) and the AI-Generated Image Naturalness (AGIN) database to further validate the generation ability of our model. AGIN aims to evaluate the naturalness of AIGC image, which is different from the IQA and IAA tasks. On these two datasets, our method also achieves impressive performance, even though UniQA is not specifically designed for them. Please refer to Appendix B.2, Table 12 and Table 13 for details.
>
> **Results on AIGIQA-20K.** The * in table indicates that we also unfreeze the backbone for training with a smaller learning rate of 2e-6, which can achieve better performance.
>
> | Method | SRCC | PLCC |
> | --- | --- | --- |
> | CLIPIQA | 0.331 | 0.483 |
> |CLIIQA+Finetune | 0.786 | 0.712|
> |CNNIQA  | 0.330 | 0.367|
> |CNNIQA  | 0.330 | 0.367|
> |Q-Align   | 0.746 | 0.742|
> |DBCNN    | 0.471 | 0.512|
> |DBCNN+Finetune    | 0.851| 0.869|
> |Ours  | 0.576 | 0.563 |
> |Ours+Finetune  | 0.830 | 0.885  |
> |Ours+Finetune* |**0.858** |**0.901** |
>
> **Results on ECIQAD.**
>
> | Method                           | SRCC   | PLCC   |
> |----------------------------------|--------|--------|
> | BRISQUE                      | 0.436  | 0.459  |
> | BIQME                       | 0.770  | 0.768  |
> | BPRI                        | 0.152  | 0.181  |
> | FRIQUEE                     | 0.663  | 0.656  |
> | CIQA                         | 0.738  | 0.735  |
> | ECIQ                        | 0.839  | 0.842  |
> | Ours                        | **0.873** | **0.887** |
> | Ours$^{*}$                   | **0.918** | **0.928** |
>
> **Results on AGIN.**
>
> | Methods           | Technical SRCC | Technical PLCC | Rationality SRCC | Rationality PLCC | Naturalness SRCC | Naturalness PLCC |
> |--------------------|----------------|----------------|-------------------|-------------------|------------------|------------------|
> | BRISQUE           | 0.4867         | 0.4909         | 0.3608           | 0.3684           | 0.3745          | 0.4067          |
> | NIQE              | 0.4235         | 0.4279         | 0.3144           | 0.3211           | 0.3358          | 0.3378          |
> | DBCNN             | 0.7623         | 0.7661         | 0.6834           | 0.6838           | 0.7053          | 0.7128          |
> | HyperIQA          | 0.7752         | 0.7806         | 0.7196           | 0.7292           | 0.7365          | 0.7509          |
> | MUSIQ             | 0.7268         | 0.7355         | 0.6916           | 0.7013           | 0.7066          | 0.7139          |
> | UNIQUE            | 0.7358         | 0.7441         | 0.6934           | 0.6976           | 0.7104          | 0.7178          |
> | MANIQA            | 0.7763         | 0.7912         | 0.7192           | 0.7217           | 0.7355          | 0.7343          |
> | PAIAA             | 0.4763         | 0.4833         | 0.4532           | 0.4536           | 0.4453          | 0.4528          |
> | TANet             | 0.5882         | 0.6143         | 0.5037           | 0.4942           | 0.4948          | 0.4815          |
> | Del. Transf.      | 0.4299         | 0.4380         | 0.4009           | 0.4016           | 0.4196          | 0.4184          |
> | SAAN              | 0.8173         | 0.8235         | 0.7564           | 0.7711           | 0.7996          | 0.8028          |
> | JOINT             | 0.8351         | 0.8429         | 0.8033           | 0.8127           | 0.8264          | 0.8362          |
> | JOINT++           | **0.8351**     | **0.8429**     | **0.8033**       | **0.8127**       | **0.8264**      | **0.8362**      |
> | Ours              | 0.7524         | 0.8007         | 0.7728           | 0.7793           | 0.7882          | 0.7979          |
> | Ours*              | **0.7785**         | **0.8104**         | 0.7898**           | **0.7952**           | **0.8069**          | **0.8171**          |

---

> ### Author Response · Authors · 2024-11-25
> **Response to Reviewer 6yW7 [3/3]**
>
> > **Q3: Is IAA a subtask of IQA, or an equally important task?**
>
> IQA and IAA are equally important, focusing on quality and aesthetics respectively. Many researchers have proposed methods to solve the two tasks independently. As you said, quality assessment carries aesthetic information. Aesthetic perception also takes image quality into account. Therefore, we aim to extract common and beneficial representations through unified pre-training, which will benefit both tasks at the same time. From the upper part of Table 6, we can notice that only using Y_IQA achieved 0.871 on CLIVE and 0.755 on AVA, which is much lower than 0.890 on CLIVE and 0.776 on AVA of joint training. Therefore, we believe that joint training of the two tasks is useful and effective.

---

> > ### Comment · Reviewer_6yW7 · 2024-12-02
> >
> > I appreciate the author's effort in responding to such many reviewers. I’ll keep my positive ratings.

---

### Official Review · Reviewer_QxF7 · 2024-11-04

**Soundness:** 2
**Presentation:** 2
**Contribution:** 2
**Rating:** 5
**Confidence:** 3

**Summary:**

This paper has two main contributions
1) This paper constructs a high-quality image-text dataset about image quality and aesthetics.
2) This paper proposes UniQA to effectively learn a general perception of image assessment.

**Strengths:**

1) This paper is well-written.
2) This paper achieves the SOTA performance on IQA and IAA tasks

**Weaknesses:**

The overall framework and concept are simple and lack novelty.
1) Directly utilizing MLLMs to generate text is now common practice.
2) The pre-training design is straightforward and typical of MLLMs.
3) This paper reports only the performance on IQA and IAA, offering a limited range of downstream tasks.

**Questions:**

Could you demonstrate the effectiveness of your design in generating high-quality text and apply UniQA to a broader range of downstream tasks ?

---

> ### Author Response · Authors · 2024-11-25
> **Response to Reviewer QxF7 [1/3]**
>
> We sincerely appreciate your helpful feedback. Your guidance is crucial in advancing our work. We have modified the paper based on your valuable comments, marked in red.
>
> > **Q1: Directly utilizing MLLMs to generate text is now common practice. The pre-training design is straightforward and typical of MLLMs.**
>
> - We would like to clarify that our motivation is to conduct unified pre-training of IQA and IAA tasks. Therefore, how to achieve effective multimodal pre-training in the fields of IQA and IAA, and how to apply the pre-trained UniQA to various image evaluation scenarios are the challenges and focuses of our work. To achieve this goal, we use MLLM to generate IQA and IAA texts and contrastive learning to pre-train our model. We propose MOS-guided task-specific prompt strategy to help MLLM generate appropriate text descriptions. The prompt strategy for image assessment is also our innovation that distinguishes it from other MLLM caption generation methods. In addition, we also proposed a data purification strategy and adapter-based lightweight fine-tuning method for high-quality multimodal pre-training and efficient downstream tasks fine-tuning, respectively.
> - This pre-trained model can be applied to various IQA and IAA tasks, including full supervision, zero-shot, few-label, and image-text retrieval, and can achieve impressive performance. In addition, UniQA can also be effectively applied to AIGC image evaluation and medical image evaluation and other realistic scenarios (detailed in Q2). This demonstrates the effectiveness of our pre-training and the generalization of UniQA.

---

> ### Author Response · Authors · 2024-11-25
> **Response to Reviewer QxF7 [2/3]**
>
> > **Q2: This paper reports only the performance on IQA and IAA, offering a limited range of downstream tasks. Could you apply UniQA to a broader range of downstream tasks?**
>
> - **Report only the performance on IQA and IAA.** Our UniQA is pre-trained on large-scale quality and aesthetics related image-text data, thus the UniQA is mainly focuses on image evaluation task. We have evaluated our model on 9 IQA and IAA datasets. In addition, we verify our model on few-label and zero-shot image assessment ability. Our model can achieve impressive results, e.g., achieving the SRCC values of 0.828 (vs. 0.764 on CLIVE of GRepQ) and 0.844 (vs. 0.812 on KonIQ of GRepQ) on few-label IQA.
> - **Apply UniQA to a broader range of downstream tasks.** To further validate our model, we supplement three other scene image evaluation tasks. These have been added to Appendix B.2, Table 11, Table 12, and Table 13, marked in red. Specifically, we use AIGC IQA datatset AIGIQA-20K, the enhanced colonoscopy image quality assessment dataset (ECIQAD) and the AI-Generated Image Naturalness (AGIN) dataset. AIGIQA-20K and ECIQAD are different image evaluation scenes from natural IQA task. AGIN aims to evaluate the naturalness of AIGC image, which is different from the IQA and IAA tasks. As shown in the table below, our model achieves highly competitive results, even though UniQA is not specifically designed for these tasks. These results further verify the generalization ability of our model.
>
> **Results on AIGIQA-20K.** The * in table indicates that we also unfreeze the backbone for training with a smaller learning rate of 2e-6, which can achieve better performance.
>
> | Method | SRCC | PLCC |
> | --- | --- | --- |
> | CLIPIQA | 0.331 | 0.483 |
> |CLIIQA+Finetune | 0.786 | 0.712|
> |CNNIQA  | 0.330 | 0.367|
> |CNNIQA  | 0.330 | 0.367|
> |Q-Align   | 0.746 | 0.742|
> |DBCNN    | 0.471 | 0.512|
> |DBCNN+Finetune    | 0.851| 0.869|
> |Ours  | 0.576 | 0.563 |
> |Ours+Finetune  | 0.830 | 0.885  |
> |Ours+Finetune* |**0.858** |**0.901** |
>
> **Results on ECIQAD.**
>
> | Method                           | SRCC   | PLCC   |
> |----------------------------------|--------|--------|
> | BRISQUE                      | 0.436  | 0.459  |
> | BIQME                       | 0.770  | 0.768  |
> | BPRI                        | 0.152  | 0.181  |
> | FRIQUEE                     | 0.663  | 0.656  |
> | CIQA                         | 0.738  | 0.735  |
> | ECIQ                        | 0.839  | 0.842  |
> | Ours                        | **0.873** | **0.887** |
> | Ours$^{*}$                   | **0.918** | **0.928** |
>
> **Results on AGIN.**
>
> | Methods           | Technical SRCC | Technical PLCC | Rationality SRCC | Rationality PLCC | Naturalness SRCC | Naturalness PLCC |
> |--------------------|----------------|----------------|-------------------|-------------------|------------------|------------------|
> | BRISQUE           | 0.4867         | 0.4909         | 0.3608           | 0.3684           | 0.3745          | 0.4067          |
> | NIQE              | 0.4235         | 0.4279         | 0.3144           | 0.3211           | 0.3358          | 0.3378          |
> | DBCNN             | 0.7623         | 0.7661         | 0.6834           | 0.6838           | 0.7053          | 0.7128          |
> | HyperIQA          | 0.7752         | 0.7806         | 0.7196           | 0.7292           | 0.7365          | 0.7509          |
> | MUSIQ             | 0.7268         | 0.7355         | 0.6916           | 0.7013           | 0.7066          | 0.7139          |
> | UNIQUE            | 0.7358         | 0.7441         | 0.6934           | 0.6976           | 0.7104          | 0.7178          |
> | MANIQA            | 0.7763         | 0.7912         | 0.7192           | 0.7217           | 0.7355          | 0.7343          |
> | PAIAA             | 0.4763         | 0.4833         | 0.4532           | 0.4536           | 0.4453          | 0.4528          |
> | TANet             | 0.5882         | 0.6143         | 0.5037           | 0.4942           | 0.4948          | 0.4815          |
> | Del. Transf.      | 0.4299         | 0.4380         | 0.4009           | 0.4016           | 0.4196          | 0.4184          |
> | SAAN              | 0.8173         | 0.8235         | 0.7564           | 0.7711           | 0.7996          | 0.8028          |
> | JOINT             | 0.8351         | 0.8429         | 0.8033           | 0.8127           | 0.8264          | 0.8362          |
> | JOINT++           | **0.8351**     | **0.8429**     | **0.8033**       | **0.8127**       | **0.8264**      | **0.8362**      |
> | Ours              | 0.7524         | 0.8007         | 0.7728           | 0.7793           | 0.7882          | 0.7979          |
> | Ours*              | **0.7785**         | **0.8104**         | 0.7898**           | **0.7952**           | **0.8069**          | **0.8171**          |

---

> ### Author Response · Authors · 2024-11-25
> **Response to Reviewer QxF7 [3/3]**
>
> > **Q3: Could you demonstrate the effectiveness of your design in generating high-quality text?**
>
> We propose a MOS-guided task-specific prompt strategy for MLLM captioning. We demonstrate the effectiveness of our strategy in Appendix D and Figure 13. Our strategy can help the model output correct text descriptions. In addition, in the ablation experiment in Table 6 (Ablation on different pre-training data), pre-training model with the text generated by MLLM ($Y_{IQA}$ and $Y_{IAA}$) can significantly enhance the performance of our model in IQA and IAA tasks. For example, using $Y_{IQA}$ improves the KonIQ from 0.907 to 0.914 SRCC and AVA from 0.748 to 0.755 SRCC; using $Y_{IAA}$ improves the KonIQ from 0.907 to 0.917 SRCC and AVA from 0.748 to 0.755 SRCC. These results prove the effectiveness of our generated text.

---

> ### Author Response · Authors · 2024-11-29
> **To Reviewer QxF7**
>
> Dear Reviewer QxF7,
>
> We truly appreciate your guidance to advance our work. We genuinely value the time and effort you dedicated to reviewing our paper. We are reaching out to ensure that our response adequately addressed all the questions and concerns you raised.
>
> Thank you for your valuable time, and we eagerly await your response.
>
> Best regards,
>
> Authors

---

> ### Author Response · Authors · 2024-12-01
> **To Reviewer QxF7**
>
> Dear Reviewer QxF7,
>
> We truly appreciate your guidance to advance our work. We genuinely value the time and effort you dedicated to reviewing our paper. Considering that the discussion will end soon, we eagerly look forward to your response.
>
> Best regards,
>
> Authors

---

> > ### Comment · Reviewer_QxF7 · 2024-12-02
> >
> > I appreciate the author's effort in responding my questions. However, the actual innovation of this paper is very limited, so  I’ll keep my rating.

---

> ### Author Response · Authors · 2024-12-02
> **Response to Reviewer QxF7**
>
> Thanks for your further response. Regarding your concerns about innovation, we have the following response:
>
> > **The innovation of this paper.**
>
> - We want to clarify that **our work focuses on unified pre-training for joint IQA and IAA tasks to benefit various image assessment tasks.** To achieve this, we use MLLM to generate text with well-designed MOS-guided task-specific prompts and use the generated text to help us with data purification. Experiments show that the pre-trained model is beneficial for both IQA and IAA tasks. This provides inspiration for future researchers to create a more unified and universal image assessment model.
> - We propose prompt strategies and data purification strategies to help MLLM generate correct text and purify data. We propose a **MOS-guided task-specific prompt** to effectively guide MLLM generate correct description. Using MOS as a condition to control LMM to generate quality-related captions is innovative and meaningful. We introduce a simple yet effective Aesthetics-relevance and Informativeness Rank (AIR) to purify data. The work on dataset construction is a highlight of this paper.
> - **Our pre-trained model can be applied to various image assessment scenarios, including full supervision, zero-shot, few-label, image-text retrieval and other downstream image assessment tasks.** For example, UniQA can be effectively applied to AIGC image quality assessment, AIGC Image Naturalness assessment, and medical image assessment and other realistic scenarios. Therefore, our model has excellent generalization ability that can have beneficial effects on other image assessment tasks.

---

> > ### Author Response · Authors · 2024-12-03
> > **To Reviewer QxF7**
> >
> > Dear Reviewer QxF7,
> >
> > We truly appreciate your guidance to advance our work. We genuinely value the time and effort you dedicated to reviewing our paper. We are reaching out to ensure that our response adequately addressed all the questions and concerns you raised.
> >
> > We will open source this large-scale AI-generated text dataset on image quality and aesthetics. We believe that this will be useful for IQA and IAA methods based on multimodal learning. In addition, our method shows excellent performance in the field of AIGC image (Table 4, Table 11 and Table 13), which is also helpful for the future field of AIGC image quality assessment. In addition, our method can also be generalized to the field of medical image assessment (Table 12). In summary, our method can contribute to the field of image assessment.
> >
> > Thank you for your valuable time, and we eagerly await your response.
> >
> > Best regards,
> >
> > Authors

---

### Official Review · Reviewer_Grpo · 2024-11-04

**Soundness:** 2
**Presentation:** 3
**Contribution:** 2
**Rating:** 5
**Confidence:** 4

**Summary:**

This paper aims to leverage unified vision-language pre-training to address quality and aesthetic assessment problems concurrently, and proposes a method named UniQA. On the one hand, this paper constructs a high-quality image-text dataset about quality and aesthetics with the assistance of MLLMs. On the other hand, UniQA learns the shared representations of IQA and IAA tasks by pre-training on the constructed dataset.  Additionally, a Multi-Cue Integration Adapter is proposed in UniQA for downstream assessment tasks.

**Strengths:**

1. In this paper, a high-quality image-text dataset about image quality and aesthetics is constructed based on the assistance of MLLMs, which is valuable．
2．The organizational structure of the article is clear and the content is complete. The writing is clear and easy to follow.
3. The motivation to "extract mutually beneficial and effective representations for both IQA and IAA tasks" in this paper is plausible.
4. This paper proposes an effective data purification strategy that refines the raw aesthetic caption dataset, providing valuable insights for data organization and cleaning in the IAA field.

**Weaknesses:**

1．The authors highlight that the motivation of this paper is to "extract mutually beneficial and effective representations for both IQA and IAA tasks." However, throughout the paper, neither the proposed dataset nor the proposed method fully explore the mutually beneficial representations for IQA and IAA tasks; instead, they only address the creation of effective representations for these tasks. Specifically, the method proposed in this paper learns a shared feature representation for IQA and IAA tasks, without proving how the representation is "mutually beneficial" which is somewhat disappointing. It is suggested to provide specific experiments or analyses that demonstrate how the learned representations benefit both IQA and IAA tasks mutually.
2．The experiments in this paper are not sufficiently comprehensive. First, the compared methods are relatively outdated, lacking comparisons with more recent works in 2024, such as [1][2]. Additionally, the ablation study only focuses on the IQA task, without any ablation analysis for the IAA task, which makes the conclusions of the ablation study less convincing.
3．The novelty is limited. The pretraining process of UniQA merely applies a standard contrastive learning strategy, a commonly used approach in numerous previous works. Additionally, the Multi-Cue Integration Adapter directly adopts the inference approach of CLIP-IQA, while the visual feature adaptation module functions similarly to a LoRA operation for fine-tuning. This Adapter appears unrelated to the concept of Multi-Cue Integration.  It is not clear how the proposed approach improves upon or differs from standard contrastive learning and existing methods like CLIP-IQA. And  a more detailed explanation of how the Adapter relates to the concept of Multi-Cue Integration is suggested.
4．This paper dedicates extensive sections to describing the dataset construction process but lacks detailed information about the resulting dataset, such as sample size, text length distribution, or word cloud distribution. There is also an absence of detailed statistical analysis and comparison between the IQA and IAA datasets used for training, leaving the relationships and differences between them unexplored.
5. The paper contains minor errors that need careful review. For instance, in Line 365, "supplementary material" should be referred to as the "Appendix."

[1] Wu H, Zhang Z, Zhang W, et al. Q-align: Teaching lmms for visual scoring via discrete text-defined levels. ICML 2024

[2] Shi T, Chen C, Wu Z, et al. Improving Image Aesthetic Assessment via Multiple Image Joint Learning. ACM Transactions on Multimedia Computing, Communications and Applications, 2024.

**Questions:**

1. In Eq. (5), simply summing AR and IR may not be the optimal approach. Assigning different weights to AR and IR and conducting relevant experiments to determine the best weight parameters could yield better results.
2. In Line 103, the statement "achieving SRCC values of 0.828 (vs. 0.764 on CLIVE) and 0.844 (vs. 0.812 on KonIQ)" is somewhat confusing, as it is unclear which method these results are being compared against.

---

> ### Author Response · Authors · 2024-11-25
> **Response to Reviewer Grpo [1/2]**
>
> We sincerely appreciate your helpful feedback. Your guidance is crucial in advancing our work. We have modified the paper based on your valuable comments, marked in red.
>
> > **Q1: Provide specific experiments or analyses that demonstrate how the learned representations benefit both IQA and IAA tasks mutually.**
>
> **Firstly**, we add the Grad-CAM of the aesthetic pre-training model in Figure 5. As shown in Figure 5, the focus of quality and aesthetics overlaps, showing the commonality between IQA and IAA. The unified pre-training of quality and aesthetics can focus on the areas of IQA and IAA tasks at the same time. Therefore, the representation of the two tasks can help the model focus on more image perception information. We have supplemented the visualization analysis of 5.6 (Impact of different pre-training data), marked in red. **Secondly**, our ablation experiments in Table 6 (Ablation on different pre-training data) also show that using the $Y_{IQA}$ dataset improves AVA (IAA task, from 0.748 to 0.755 SRCC), and using the $Y_{IAA}$ dataset improves KonIQ (IQA task, from 0.907 to 0.917). This shows that the two tasks are mutually beneficial. When we train the IQA and IAA data together, the model achieves the best results.
>
> ---
>
> > **Q2: The compared methods are relatively outdated, lacking comparisons with more recent works in 2024. The ablation study only focuses on the IQA task, without any ablation analysis for the IAA task.**
>
> - **Compared methods.** As you suggested, we have added recent methods for comparison [1-2], as shown in Table 1 and Table 2. In addition, for a more comprehensive comparison, we added more comparison methods [3-5] in Table 15 in the Appendix.
> - **Ablation analysis for the IAA task.** Regarding the ablation experiment, the reviewer may have a misunderstanding. As shown in Table 6, we have used the classic IAA dataset AVA for ablation study. When ablating modules, we will analyze by referring to the performance of both IQA and IAA tasks.
>
> [1] Wu H, Zhang Z, Zhang W, et al. Q-align: Teaching lmms for visual scoring via discrete text-defined levels. ICML 2024
>
> [2] Shi T, Chen C, Wu Z, et al. Improving Image Aesthetic Assessment via Multiple Image Joint Learning. ACM Transactions on Multimedia Computing, Communications and Applications, 2024.
>
> [3] Zhong Y, Wu X, Zhang L, et al. Causal-IQA: Towards the Generalization of Image Quality Assessment Based on Causal Inference[C]//Forty-first International Conference on Machine Learning.2024
>
> [4] Avanaki N J, Ghildiyal A, Barman N, et al. LAR-IQA: A Lightweight, Accurate, and Robust No-Reference Image Quality Assessment Model[J]. arXiv preprint arXiv:2408.17057, 2024.
>
> [5] Fu H, Wang Y, Yang W, et al. DP-IQA: Utilizing Diffusion Prior for Blind Image Quality Assessment in the Wild[J]. arXiv preprint arXiv:2405.19996, 2024.
>
> ---
>
> > **Q3: Difference with standard contrastive learning and existing methods like CLIP-IQA.**
>
> - **Difference with standard contrastive learning.** We use the same contrastive learning approach as CLIP. Note that our work focuses on unified pre-training for joint IQA and IAA tasks. To achieve this, we propose MOS-guided task-specific prompts to guide MLLM generate text, and Aesthetics-relevance and Informativeness Rank (AIR) to help us with data purification. CLIP and contrastive learning are classic multimodal model and pre-training method, so we follow this pre-training pipeline.
>
> - **Difference with CLIP-IQA.** There are two differences between our method and CLIPIQA: 1) We use a visual adapter to adjust visual features, while CLIPIQA uses CoOp to adjust input embedding. 2) CLIPIQA only uses "Good photo" and "Bad photo" for fine-tuning, and we use more prompts/cues, i.e.,{bad, poor, fair, good, perfect}. This strategy helps the model evaluate the image more comprehensively. The ablation experiment in Table 6 (Ablation on the proposed adapter) shows that compared with using a single prompt ("good image", 0.75 SRCC on CLIVE) and Antonym Prompt ("good image" and "bad image", 0.875 SRCC on CLIVE), our method can achieve better results (five level prompts/cues, 0.890 SRCC on CLIVE).
>
> ---
>
> >**Q4: How the Adapter relates to the concept of Multi-Cue Integration?**
>
> The “multi-cue” means that we use more prompts to evaluate an image, that is, {bad, poor, fair, good, perfect}. CLIPIQA [1] propose to use "good image" and "bad image" as anchors for multimodal image evaluation. Our proposed adapter uses 5 levels (i.e., multi-cue) of prompts to more comprehensively evaluate image quality. We have modified Section 4.1 to further clearly indicate the meaning of “multi-cue”, marked in red.
>
> [1] Wang J, Chan K C K, Loy C C. Exploring clip for assessing the look and feel of images[C]//Proceedings of the AAAI Conference on Artificial Intelligence. 2023, 37(2): 2555-2563.

---

> ### Author Response · Authors · 2024-11-26
> **Response to Reviewer Grpo [2/2]**
>
> > **Q5: lacks detailed information about the resulting dataset.**
>
> Thank you for your constructive suggestions. We provide a detailed introduction to the dataset in Appendix C, including the data volume of IQA and IAA, the length distribution of text sentences, and word clouds. In summary, we use FLIVE (IQA dataset, 39,807 images) and AVA (IAA dataset, 234,090 images) with a total of 1,240,915 text descriptions. We generate three captions for each IQA image and one caption for each IAA image, resulting 119,421 generated IQA captions and 234,090 IAA captions. The text length is concentrated in 20-30 words. The word cloud shows that it can be seen that the most common words in the text dataset are aesthetic and quality-related words, such as “aesthetics”, “quality”, “composition”, etc. This indicates that the text of the constructed dataset focuses on image assessment. Please refer to Appendix C (marked in red) and Figures 6, 7, and 8 for details.
>
> ---
>
> > **Q6: The paper contains minor errors that need careful review.**
>
> We have corrected this error and re-reviewed the entire paper carefully.
>
> ---
>
> > **Q7: Assigning different weights to AR and IR.**
>
> In fact, we have discussed this strategy in Appendix A.1. We think it is more reasonable and effective method to use different factors to weight AR and IR. Considering the simplicity, we set the two factors to 1. From the Table 6 (Ablation on data purification strategy), we can notice that our strategy can bring performance improvements to the model, e.g., 0.876 to 0.890 SRCC on LIVEC with our AIR  strategy. In the future, if we collect more data, we will discuss how to assign the weights of the two strategies.
>
> ---
>
> > **Q8: In Line 103, the statement is somewhat confusing.**
>
> Thank you for your help in making our paper clearer. We have corrected the text to "achieving SRCC values of 0.828 (vs. 0.760 on CLIVE of GRepQ) and 0.844 (vs. 0.812 on KonIQ of GRepQ)". We have corrected the PDF and marked it in red.

---

> ### Author Response · Authors · 2024-11-29
> **To Reviewer Grpo**
>
> Dear Reviewer Grpo,
>
> We truly appreciate your guidance to advance our work. We genuinely value the time and effort you dedicated to reviewing our paper. We are reaching out to ensure that our response adequately addressed all the questions and concerns you raised.
>
> Thank you for your valuable time, and we eagerly await your response.
>
> Best regards,
>
> Authors

---

> > ### Comment · Reviewer_Grpo · 2024-12-02
> >
> > Thank you for the reply. However, my concerns have not been adequately addressed. Specifically, regarding Q4, as the authors state, "The adapter in this paper employs 5 levels (i.e., multi-cue) of prompts to more comprehensively evaluate image quality, which is termed Multi-Cue Integration." I do not consider this a novel contribution, as the method has already been proposed in the LIQE [1]. Furthermore, concerning Q3, the use of more prompts/cues, namely {bad, poor, fair, good, perfect}, for fine-tuning in this paper is not a fundamental difference from CLIP-IQA [2]. In fact, CLIP-IQA [2] has already experimented with various prompts. Therefore, on the whole, the novelty of this paper is limited.
> >
> > [1] Zhang W, Zhai G, Wei Y, et al. Blind image quality assessment via vision-language correspondence: A multitask learning perspective[C]//Proceedings of the IEEE/CVF conference on computer vision and pattern recognition. 2023: 14071-14081.
> >
> > [2] Wang J, Chan K C K, Loy C C. Exploring clip for assessing the look and feel of images[C]//Proceedings of the AAAI Conference on Artificial Intelligence. 2023, 37(2): 2555-2563.

---

> ### Author Response · Authors · 2024-12-01
> **To Reviewer Grpo**
>
> Dear Reviewer Grpo,
>
> We truly appreciate your guidance to advance our work. We genuinely value the time and effort you dedicated to reviewing our paper. Considering that the discussion will end soon, we eagerly look forward to your response.
>
> Best regards,
>
> Authors

---

> ### Author Response · Authors · 2024-12-02
> **Response to Reviewer Grpo**
>
> We thank the reviewer for his further meaningful response. Regarding the novelty and differences between this paper and LIQE and CLIPIQA, we have the following responses:
>
> > **Differences with LIQE and CLIPIQA.**
>
> - **Differences with LIQE**. LIQE uses five levels of prompts for quality evaluation. However, LIQE uses the entire CLIP for fine-tuning, while we only use a lightweight adapter. Our adapter has only 0.26M learnable parameters, while LIQE has 151M parameters. We have reported the comparative results in Table 1. Our method has superior performance, such as 0.963 SRCC (vs 0.936 of LIQE) on CSIQ and 0.933 SRCC (vs. 0.919 of LIQE) on KonIQ.
>
> - **Differences with CLIPIQA**. CLIPIQA discusses the effects of different prompt templates and different prompts on the model performance. However, it does not use the prompt ensemble strategy (using multiple prompts at the same time and take the average score as final score). We find that the prompt ensemble strategy has a significant effect on zero-shot (Table 4) and few-label (Table 5) image evaluation scenarios. The excellent performance in the few-label scenario is a highlight of our article.
>
> Despite the above differences, we still want to emphasize that the innovation of this paper lies in the multimodal pre-training of image assessment, as detailed below.
>
> > **Novelty of the paper.**
>
> - We want to clarify that **our work focuses on unified pre-training for joint IQA and IAA tasks to benefit various image assessment tasks.** To achieve this, we use MLLM to generate text with well-designed MOS-guided task-specific prompts and use the generated text to help us with data purification. Experiments show that the pre-trained model is beneficial for both IQA and IAA tasks. This provides inspiration for future researchers to create a more unified and universal image assessment model.
> - We propose prompt strategies and data purification strategies to help MLLM generate correct text and purify data. We propose a **MOS-guided task-specific prompt** to effectively guide MLLM generate correct description. Using MOS as a condition to control LMM to generate quality-related captions is innovative and meaningful. We introduce a simple yet effective Aesthetics-relevance and Informativeness Rank (AIR) to purify data. The work on dataset construction is a highlight of this paper.
> - **Our pre-trained model can be applied to various image assessment scenarios, including full supervision, zero-shot, few-label, image-text retrieval and other downstream image assessment tasks.** For example, UniQA can be effectively applied to AIGC image quality assessment, AIGC Image Naturalness assessment, and medical image assessment and other realistic scenarios. Therefore, our model has excellent generalization ability that can have beneficial effects on other image assessment tasks.

---

> ### Author Response · Authors · 2024-12-03
> **Response to Reviewer Grpo**
>
> Dear Reviewer Grpo,
>
> We truly appreciate your guidance to advance our work. We genuinely value the time and effort you dedicated to reviewing our paper. We are reaching out to ensure that our response adequately addressed all the questions and concerns you raised.
>
> We will open source this large-scale AI-generated text dataset on image quality and aesthetics. We believe that this will be useful for IQA and IAA methods based on multimodal learning. In addition, our method shows excellent performance in the field of AIGC image (Table 4, Table 11 and Table 13), which is also helpful for the future field of AIGC image quality assessment. In addition, our method can also be generalized to the field of medical image assessment (Table 12). In summary, our method can contribute to the field of image assessment.
>
> Thank you for your valuable time, and we eagerly await your response.
>
> Best regards,
>
> Authors

---

### Official Review · Reviewer_r5Qs · 2024-11-04

**Soundness:** 2
**Presentation:** 3
**Contribution:** 3
**Rating:** 8
**Confidence:** 5

**Summary:**

This paper proposes a unified evaluation model to handle both image quality assessment (IQA) and image aesthetic assessment (IAA) tasks simultaneously. Specifically, the proposed method first leverages the existing multimodal large language model (MLLM) to generate IQA (YIQA), IAA (YIAA) datasets with generated descriptions, and purify IAA dataset (Y+IAA) annotated by human’s comments. Then, the proposed method train the CLIP model based on YIQA, YIAA and Y+IAA. Finally, an multi-cue integration adapter is designed to allow the pre-trained CLIP to adapt to specific datasets.

**Strengths:**

1. This paper includes comprehensive datasets, encompassing nearly all existing IQA and IAA datasets, providing robust validation for the effectiveness of the proposed method.
2. The methodology of the algorithm is described in a clear and straightforward manner, with concise and easily understandable language.
3. The paper presents an extensive set of experiments and rich visualizations, thoroughly validating each module within the algorithm's design.

**Weaknesses:**

1. The paper’s motivation appears to lack practical significance or has not been convincingly demonstrated.
2. While the proposed approach is intricate, its actual innovation is minimal.
3. The related work section includes methods that are either outdated or lack representativeness in the current IQA and IAA research.
4. Beyond metric improvements, the proposed method lacks substantial inspirational value for future studies.
5. The paper is missing some essential experiments that could substantiate its motivations.

**Questions:**

1. Motivation Concerns: The motivation for jointly considering IQA and IAA tasks could be questioned. Is there a urgent or practical need to address IQA and IAA together? Do these tasks mutually benefit each other, or does their combination offer real-world value beyond mere metric improvements?
2. Combined Dataset Training: The authors claim that existing combined dataset training fails to learn mutually beneficial representations shared by both tasks. In fact, the authors construct three datasets (YIQA, YIAA and Y+IAA) to train a CLIP model, which resembles the challenge they raised. This raises questions about the rationality of their motivation and calls for concrete evidence to support it.
3. Noise in human- annotated IAA Datasets: The authors argue that textual noise in IAA datasets negatively impact prediction performance. It would be more convincing if they specified what constitutes " textual noise " and clarified the specific negative impacts. In addition, the claim that MLLM-generated description can purify " textual noise " is unconvincing, as only one MLLM model was used in description generation. The model’s biases and possible overuse of irrelevant words are not addressed, making this assumption lack strong justification.
4. Common feature representation: The authors suggest that the proposed method can extract common representations from IQA and IAA tasks. Visual evidence supporting this claim would strengthen their argument.
5. Informativeness Rank (IR): Using sentence length as the metric for Informativeness Rank might be biased. The number of irrelevant or potentially harmful words within a sentence should not be overlooked.
6. Related Work: The IQA and IAA methods discussed in the related work section are relatively outdated. Adding more recent and representative algorithms would improve this section.
7. Experimental Comparisons and Visualization Limitations: Some newest comparison methods are required. Additionally, the visualization in Fig. 5 fails to demonstrate that the proposed method focuses more on noisy objects and backgrounds. The examples tend to provide “blurry image”, but the sample mostly depicts a blurred background with clear objects, which does not align with the stated objective.

---

> ### Author Response · Authors · 2024-11-26
> **Response to Reviewer r5Qs [1/3]**
>
> Thank you very much for your suggestions! We sincerely hope our response can help address your concerns. If you have any other questions, we would be more than happy to respond !
>
> > **Q1: The paper’s motivation appears to lack practical significance or has not been convincingly demonstrated. Beyond metric improvements, the proposed method lacks substantial inspirational value for future studies. Is there a urgent or practical need to address IQA and IAA together? Do these tasks mutually benefit each other, or does their combination offer real-world value beyond mere metric improvements?**
>
> - **Necessity**: Joint training is helpful for learning human perception of images and usage in real scenes. Specifically, because both IQA and IAA focus on image evaluation tasks, jointly training the two tasks can learn human perceptual representations of images. Secondly, in some real scenarios, such as using image evaluation models to select high-quality data for AIGC model training, the evaluation model needs to be able to consider both the quality and aesthetics of the image.
>
> - **Effectiveness**: The data from both tasks are mutually beneficial. From the ablation experiment in Table 6 (Ablation on different pre-training data), we can see that pre-training with Y_{IQA} data improve AVA (IAA task) performance (0.748 to 0.755 SRCC), pre-training with Y_{IAA} data improve KonIQ (IQA task) performance (0.907 to 0.917 SRCC). Therefore, the data from two tasks are mutually beneficial. When jointly trained, the performance improvement is more obvious, e.g., 0.865 to 0.890 on CLIVE (IQA task) and 0.748 to 0.776 on AVA (IAA task).
>
> - **Impact on reality**: **Firstly**, our method has excellent performance in the few-label scenario, showing that UniQA has promising prospects in helping to reduce the annotation requirements and costs. In few-label IQA, we have achieved a significant improvement, achieving the SRCC values of 0.828 (vs. 0.760 on CLIVE of GRepQ) and 0.844 (vs. 0.812 on KonIQ of GRepQ). **Secondly**, our method can be used as a foundation model and generalizes well to many IQA datasets. To further validate our model, we supplement three other scene image evaluation tasks. These have been added to the pdf and are marked in red. Please refer to Appendix B.2, Table 11, Table 12, and Table 13. Specifically, we use AIGC IQA datatset AIGIQA-20K, the enhanced colonoscopy image quality assessment dataset (ECIQAD) and the AI-Generated Image Naturalness (AGIN) dataset. We achieve highly competitive results on all three datasets. The strong generalization ability shows that UniQA can play an important role in helping image assessment tasks in other fields. **Thirdly**, many scenarios need to consider both image quality and aesthetics, such as image recommendation systems and data filtering. Our work provides inspiration for more general image evaluation systems in the future.
>
> - Reviewers Grpo, 6yW7, uVpG, and WcRe support the role and significance of our joint training.
>
> ---
>
> > **Q2: Innovation is minimal.**
>
> - In this paper, we propose unified pre-training of quality and aesthetics. Experiments show that the pre-trained model is beneficial for both IQA and IAA tasks. This provides inspiration for future researchers to create a more unified and universal image assessment model.
> - We propose prompt strategies and data purification strategies to help MLLM generate correct text and purify data. We propose a MOS-guided task-specific prompt to effectively guide MLLM generate correct description. We introduce a simple yet effective Aesthetics-relevance and Informativeness Rank (AIR) to purify data. The work on dataset construction is a highlight of this paper.
> - Our pre-trained model can be applied to various image evaluation scenarios, including full supervision, zero-shot, few-label, and image-text retrieval. In addition, UniQA can also be applied to AIGC image evaluation and medical image evaluation and other realistic scenarios. Therefore, our model has excellent generalization ability.

---

> ### Author Response · Authors · 2024-11-26
> **Response to Reviewer r5Qs [2/3]**
>
> > **Q3: Related work section includes methods that are either outdated or lack representativeness.**
>
> Thank you for your suggestions to improve our article. We have added recently published articles in related work, including IQA method[1-4] and IAA method[5-7]. We discuss the differences between these approaches and ours. We marked them in red in the article.
>
> [1] Xu K, Liao L, Xiao J, et al. Boosting Image Quality Assessment through Efficient Transformer Adaptation with Local Feature Enhancement[C]//Proceedings of the IEEE/CVF Conference on Computer Vision and Pattern Recognition. 2024: 2662-2672.
>
> [2] Shin N H, Lee S H, Kim C S. Blind Image Quality Assessment Based on Geometric Order Learning[C]//Proceedings of the IEEE/CVF Conference on Computer Vision and Pattern Recognition. 2024: 12799-12808.
>
> [3] Saha A, Mishra S, Bovik A C. Re-iqa: Unsupervised learning for image quality assessment in the wild[C]//Proceedings of the IEEE/CVF conference on computer vision and pattern recognition. 2023: 5846-5855.
>
> [4] Wu H, Zhang Z, Zhang W, et al. Q-align: Teaching lmms for visual scoring via discrete text-defined levels[J]. arXiv preprint arXiv:2312.17090, 2023.
>
> [5] Nie X, Hu B, Gao X, et al. BMI-Net: A Brain-inspired Multimodal Interaction Network for Image Aesthetic Assessment[C]//Proceedings of the 31st ACM International Conference on Multimedia. 2023: 5514-5522.
>
> [6] Huang Y, Li L, Chen P, et al. Coarse-to-fine Image Aesthetics Assessment With Dynamic Attribute Selection[J]. IEEE Transactions on Multimedia, 2024.
>
> [7] He S, Ming A, Zheng S, et al. Eat: An enhancer for aesthetics-oriented transformers[C]//Proceedings of the 31st ACM International Conference on Multimedia. 2023: 1023-1032.
>
> ---
>
> > **Q4: Missing some essential experiments that could substantiate its motivations.**
>
> We verify our motivation through the ablation experiments in Table 6. Results in Table 6 (Ablation on different pre-training data) show that using the YIQA dataset improves AVA (IAA task, from 0.748 to 0.755), and using the YIAA dataset improves KonIQ (IQA task, from 0.907 to 0.917). This shows that the two tasks are mutually beneficial. When we train the IQA and IAA data together, the model achieves the best results. These experiments validate the mutual beneficialness of IQA and IAA and the effectiveness of unified multimodal pre-training, which is our motivation.
>
> ---
>
> > **Q5: Difference with other combined dataset training methods.**
>
> Some existing joint training methods, such as Q-Align, directly combine various data sets for regression training. Q-Align's experiments show that joint training will not bring significant performance improvements. For example, training alone on SPAQ is 0.930, joint training (SPAQ+KonIQA+KADID) is 0.931. **Our method is a paradigm of pre-training plus fine-tuning.** We first conduct multi-modal pre-training on the data sets of the two tasks, and then apply the pre-trained model to other data sets through regression training through the adapter. As we know from Table 6, our pre-training can significantly improve model performance. We also compare our method with Q-Align in Table 1. Our method achieves better results on most datasets. **In addition**, our pre-trained UniQA can support more tasks compared to other unified methods. UniQA can support zero-shot, few-label, text-image retrieval and other scene (e.g., Medical image evaluation and AIGC image naturalness evaluation, details are in Table 12 and 13) image assessment tasks, with a wider range of applications.

---

> ### Author Response · Authors · 2024-11-26
> **Response to Reviewer r5Qs [3/3]**
>
> > **Q6: What is the "textual noise" and the specific negative impacts of them? Only one MLLM model was used in description generation, which may introduce bias and overuse of irrelevant words.**
>
> - **Textual noise and negative impacts.** The IAA text dataset is human comments on the website. As shown in Figure 14 in the Appendix, there are many texts in the comments that are not related to aesthetics, such as "Thanks for all your comments!". These texts are not conducive to the image-text alignment of multimodal pre-training, which has been discussed in previous work. Through our strategy, such irrelevant comments can be filtered out. Our ablation experiment in Table 6 (Ablation on data purification strategy) shows that after using our strategy, the performance of the model has improved, for example, 0.876 to 0.890 SRCC in CLIVE.
> - **MLLM's bias.** Using more MLLM can only improve the diversity of text and further improve the effect of pre-training, rather than removing irrelevant words. We have proposed the MOS-guided task-specific prompts, which include image quality guidance and assessment content guide, to help MLLM generate high-quality text and reduce hallucinations. As shown in Table 6 (Ablation on different pre-training data), using generated text can significantly improve the performance of the model, for example, 0.907 to 0.917 on the KonIQ dataset when using the YIAA dataset. Therefore, the generated text is effective and beneficial, thus helping us filter irrelevant and low-quality data.
>
> ---
>
> > **Q7: Visual evidence for supporting Common feature representation.**
>
> Thank you for your suggestion. We have added the Grad-CAM of the aesthetic pre-training model in Figure 5. As shown in Figure 5, the focus of quality and aesthetics overlaps, showing the commonality between IQA and IAA. The unified pre-training of quality and aesthetics can focus on the areas of IQA and IAA tasks at the same time. This shows that unified training can learn common representations of the two tasks. We have added these discussions to Section 5.6 and marked them in red.
>
> ---
>
> > **Q8: Using sentence length as the metric for Informativeness Rank might be biased.**
>
> Our approach has incorporated the Aesthetics-relevance Ranking, an effective metric for assessing both the quality of the text and its alignment with images. Consequently, we chose a straightforward method to reflect the text's informativeness by its sentence length. In fact, we have delved into a superior and more rational technique in Appendix A.1, where we propose varying the weights assigned to Aesthetics-relevance (AR) and Image-relevance (IR) to better understand their influence on data quality in pre-training. For simplicity, we set the two weight factors to  1. The results from the ablation study in Table 6 (Ablation on data purification strategy) demonstrate that our strategy enhances model performance. In the future, if we collect more data, we will discuss how to assign the weights of the two strategies to further improve pre-training performance. More effective ways of measuring information are also worth exploring.
>
> ---
>
> > **Q9: Experimental Comparisons and Visualization Limitations.**
>
> - Experimental Comparisons: We have added more comparison methods to Table 1 and Table 15 (which we added here due to space limitations), including Q-Align [1], CIS [2], LAR-IQA [3], and DP-IQA [4].
>
> - Visualization Limitations: We added a column of visualizations to Figure 5 where the model pays attention to both the blurred background and the object. In addition, in columns 3 and 5, the model also pays attention to the blurred object.
>
> [1] Wu H, Zhang Z, Zhang W, et al. Q-align: Teaching lmms for visual scoring via discrete text-defined levels[J]. arXiv preprint arXiv:2312.17090, 2023.
>
> [2] Zhong Y, Wu X, Zhang L, et al. Causal-IQA: Towards the Generalization of Image Quality Assessment Based on Causal Inference[C]//Forty-first International Conference on Machine Learning.2024
>
> [3] Avanaki N J, Ghildiyal A, Barman N, et al. LAR-IQA: A Lightweight, Accurate, and Robust No-Reference Image Quality Assessment Model[J]. arXiv preprint arXiv:2408.17057, 2024.
>
> [4] Fu H, Wang Y, Yang W, et al. DP-IQA: Utilizing Diffusion Prior for Blind Image Quality Assessment in the Wild[J]. arXiv preprint arXiv:2405.19996, 2024.

---

> ### Author Response · Authors · 2024-11-29
> **To Reviewer r5Qs**
>
> Thanks for raising the score. We truly appreciate your recognition of our work and are very happy to have addressed your concerns, which encourages us a lot. Best wishes.

---

### Note · Authors · 2025-01-23

I have read and agree with the venue's withdrawal policy on behalf of myself and my co-authors.